# Randomized Advantage Transformation (RAT): Computing Natural Policy Gradients via Direct Backpropagation

**Mingfei Sun** [1]

## Abstract

Natural policy gradients improve optimization by accounting for the geometry of distribution space, but their practical use is limited by the cost of estimating and inverting the Fisher matrix. We present Randomized Advantage Transformation (RAT), a method for estimating Tikhonov-regularized natural policy gradients via direct backpropagation. By applying the Woodbury formula, we reformulate the regularized natural policy gradients as vanilla policy gradients with a transformed advantage. RAT computes this transformation efficiently via randomized block Kaczmarz iterations on on-policy mini-batches, avoiding explicit Fisher construction, conjugate-gradient solvers, and architecture-specific approximations. We provide convergence guarantees for RAT and demonstrate empirically that it matches or exceeds established natural-gradient methods across continuous and visual control benchmarks, while remaining simple to implement and compatible with various architectures.

## 1. Introduction

Natural policy gradients are a foundational tool in deep Reinforcement Learning (RL), offering parameterization-invariant update directions (Bagnell & Schneider, 2003) by pre-conditioning policy gradients with the inverse Fisher matrix (Amari, 1998; Kakade, 2001). This geometric correction has been shown to significantly improve convergence properties (Agarwal et al., 2021), and underlies several influential RL algorithms, including Natural Actor-Critic (NAC) (Peters & Schaal, 2008), Trust Region Policy Optimization (TRPO) (Schulman et al., 2015), ACKTR (Wu et al., 2017), and connections to PPO-style updates (Schulman et al., 2017b; Hilton et al., 2022).

[1]The University of Manchester, United Kingdom. Code URL. Correspondence to: <mingfei.sun@manchester.ac.uk>.

*Proceedings of the 43rd International Conference on Machine Learning*, Seoul, South Korea. PMLR 306, 2026. Copyright 2026 by the author(s).

Despite their advantages, natural policy gradients are rarely used directly in large-scale deep reinforcement learning due to computational constraints. The Fisher matrix scales with the number of policy parameters, making explicit construction or inversion infeasible. To address this, prior work has largely followed two directions. Hessian-free approaches compute Fisher-vector products and solve the resulting linear systems using conjugate gradient methods, as in TRPO (Schulman et al., 2015). While effective, these methods introduce substantial computational overhead, require careful tuning of inner-loop solvers, and are difficult to apply in shared actor-critic architectures (Schulman et al., 2017b; Wu et al., 2017). Alternatively, structured approximations such as KFAC (Martens & Grosse, 2015) exploit layer-wise factorizations of the Fisher matrix, trading accuracy for efficiency but relying on architecture-dependent assumptions (Benzing, 2022) and nontrivial implementation optimizations (George et al., 2018).

In this work, we show that estimating natural policy gradients can be reduced to a simpler and more general procedure. Our starting point is the observation that Tikhonov-regularized natural policy gradients admit an equivalent least-squares formulation. By applying the Woodbury formula, we derive a representation in which the inverse Fisher matrix is absorbed into a transformation of the advantage function. Under this reformulation, the natural policy gradient takes the same form as a vanilla policy gradient, but with a modified advantage, see Figure 1 for an overview.

Building on this insight, we introduce Randomized Advantage Transformation (RAT), an algorithm that approximates the transformed advantage using randomized block Kaczmarz iterations (Needell & Tropp, 2014). RAT operates on small subsets of on-policy samples and iteratively refines an estimate of the regularized natural policy gradient. Each iteration requires only standard backpropagation through a surrogate loss, eliminating the need for explicit Fisher construction, Fisher-vector products, or architecture-specific curvature approximations.

We provide a convergence analysis showing that RAT converges linearly to the regularized natural policy gradient under standard assumptions on the function approximation and state-action coverage. We evaluate RAT on a range

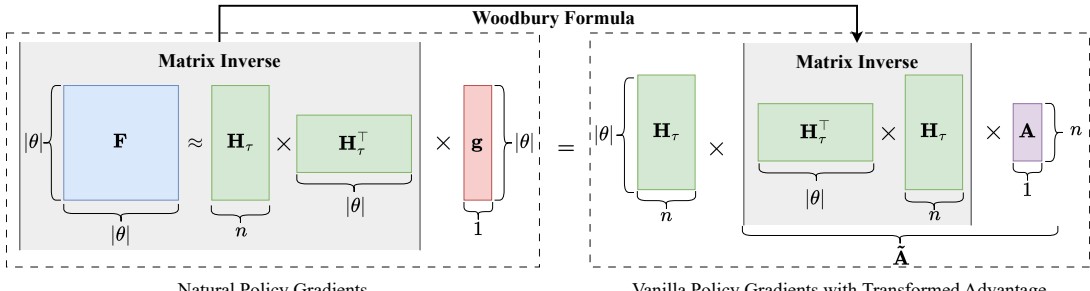

*Figure 1.* Fisher matrix $\boldsymbol{F} \in \mathbb{R}^{|\boldsymbol{\theta}| \times |\boldsymbol{\theta}|}$ estimated from samples $\tau$ (shown in green) is often ill-conditioned and hard to invert directly. Randomized Advantage Transformation (RAT) leverages Woodbury formula to replace the inversion of $\boldsymbol{F}$ to that of a sampled preconditioner of size $n \times n$ (shown in grey). The resulting inverse is absorbed into a randomized, block-wise transformation of the advantage function, yielding a surrogate objective whose first-order gradient via direct backpropagation approximates natural policy gradient.

of benchmark reinforcement learning tasks, including continuous control in MuJoCo (Brockman et al., 2016) and high-dimensional visual control in Procgen (Cobbe et al., 2020). Across these settings, RAT matches or outperforms established natural policy gradient methods while offering a simpler implementation and broader applicability, including support for shared actor-critic architectures. Our contributions are as follows:

- We derive a Woodbury-based reformulation of Tikhonov-regularized natural policy gradients as vanilla policy gradients with transformed advantages.
- We propose Randomized Advantage Transformation (RAT), an efficient algorithm based on randomized block Kaczmarz iterations that estimates NPG using standard backpropagation.
- We provide theoretical convergence guarantees and empirical evidence demonstrating the effectiveness of RAT on diverse benchmarks.

## 2. Related work

We review prior work on estimating natural policy gradients and on using the Woodbury formula to reduce the cost of Fisher inversion.

**Estimating natural policy gradients.** The central computational challenge in natural policy gradient methods is to *estimate or apply the inverse Fisher matrix* efficiently and robustly in the presence of sampling noise and function approximation. A widely adopted strategy avoids forming the Fisher explicitly and instead computes *Fisher Vector Products* (FVP), also known as Hessian-free methods (Martens, 2010; Pascanu & Bengio, 2014). The resulting linear systems are typically solved using iterative methods such as conjugate gradient (CG) algorithm. This approach underlies trust-region style reinforcement learning algorithms (Schulman et al., 2015) and many modern natural policy gradient implementations (Kuba et al., 2022; Sun et al., 2023). In

practice, however, achieving sufficient accuracy often requires many CG iterations at each training iteration, leading to substantial computational overhead. To address this issue, alternative approaches rely on *structured approximations* of the Fisher matrix, including diagonal (Liu et al., 2024), layer-wise block-diagonal (Martens & Grosse, 2015; George et al., 2018), or low-rank forms (Dangel et al., 2023; Yang et al., 2022). Among these, Kronecker-Factored Approximate Curvature (KFAC) has been widely used in reinforcement learning (Wu et al., 2017; Bae et al., 2022). KFAC approximates each layer's Fisher matrix as a Kronecker product of two smaller matrices, relying on independence assumptions about the statistics of the gradients (Ba et al., 2017) or their eigen-structure (George et al., 2018). While effective, KFAC depends strongly on the gradient structure of the neural networks. In contrast, our method RAT is oblivious to the particular model architecture and estimates natural policy gradients through standard backpropagation. Another influential line of work estimates natural policy gradients via *natural actor-critic* (NAC) formulations (Peters & Schaal, 2008; Cayci & Eryilmaz, 2025). In these methods, the advantage function is represented in a *compatible* form (Sutton et al., 1999), ensuring that the natural policy gradient emerges exactly as the solution to a least-squares regression problem (Kakade, 2001; Schulman et al., 2017a). Our work is related in spirit, but differs in how the least-squares problem is constructed and solved.

**Woodbury for inverting Fisher.** The Woodbury formula provides an efficient way to compute matrix inverses and solve linear systems involving low-rank updates. Its use in natural gradients dates back to Amari et al. (2000), where the Fisher inverse can be stored explicitly and is estimated directly using the Sherman-Morrison lemma, a special case of the Woodbury formula. More recently, this idea has been extended to rank-1 approximations of natural policy gradients with neural parameterization (Huo et al., 2026). Woodbury formula has also been used to reduce the computational cost

of natural gradient optimization steps. For example, Chen & Heyl (2024) apply Woodbury-based updates to accelerate natural gradient methods, and related work develops a momentum scheme to further improve convergence, e.g., SRPING (Goldshlager et al., 2024). Only very recently has the full Woodbury formula been used to reformulate the *Tikhonov-regularized natural gradients*. For instance, Wu et al. (2024) employs the push-through identity to analyze per-sample loss reduction under natural gradient updates, while Guzmán-Cordero et al. (2025) uses Woodbury-based transformations to reduce the cost of Fisher inverse, for which convergence guarantees have been provided in Goldshlager et al. (2026). Our approach builds on these insights but differs in how the Woodbury reformulation is exploited. Instead of directly approximating the inverse Fisher, we use the Woodbury identity to transform the advantage function and then estimate the resulting natural policy gradient iteratively through standard backpropagation.

**Position of RAT.**  RAT differs from prior natural gradient methods in that it neither relies on conjugate-gradient solvers nor on structured Fisher approximations. Instead, it leverages a Woodbury-based reformulation to shift curvature information into an advantage transformation, which is approximated via randomized linear solvers. This perspective allows RAT to remain architecture-agnostic, computationally efficient, and compatible with various architectures.

## 3. Preliminaries

**Reinforcement learning formulation.**  We consider a standard reinforcement learning setup in which an agent interacts with an environment over discrete timesteps. At each timestep $t$, the agent observes a state $s_t \in \mathcal{S}$, selects an action $a_t \in \mathcal{A}$ and receives a scalar reward $r_t \in \mathbb{R}$. The agent's behavior is defined by a stochastic policy $\pi$, which maps states to action distributions $\pi : \mathcal{S} \mapsto \Delta_{|\mathcal{A}|}$ (where $\Delta_d := \{\boldsymbol{x} \in \mathbb{R}_+^n : \sum_i^n x_i = 1\}$ denotes the probability simplex). The environment is modelled as a Markov Decision Process (MDP) $\{\mathcal{S}, \mathcal{A}, P, r, p_0\}$, where $P(\cdot|s, a)$ is the transition kernel, $r(s, a)$ is the reward function, and $p_0$ is the initial state distribution. The (discounted) return from time step $t$ is defined as $R_t := \sum_{i=t}^{T} \gamma^{i-t} r(s_i, a_i)$, where $\gamma \in [0, 1)$ is the discount factor. The objective is to learn a policy that maximizes the expected return from the initial state distribution. The action-value function of a policy $\pi$ is defined as $Q_\pi(s_t, a_t) := \mathbb{E}[R_t | s_t, a_t]$. The value function $V_\pi$ and advantage function $A_\pi(s, a)$ are given by: $V_\pi(s_t) := \mathbb{E}_{a_t \sim \pi(\cdot|s_t)}[Q_\pi(s_t, a_t)]$, and $A_\pi(s, a) := Q_\pi(s, a) - V_\pi(s)$. We define the discounted state distribution as $d_\pi(s) := (1 - \gamma) \sum_{t=0}^{\infty} \gamma^t P(s_t = s | \pi, p_0)$, where the factor $(1 - \gamma)$ ensures normalization. With slight abuse of notation, we use $d_\pi(s, a)$ to mean $s \sim d_\pi(s)$, $a \sim \pi(\cdot|s)$.

**Vanilla and Natural Policy Gradients.**  We consider a parameterized policy $\pi(a|s; \boldsymbol{\theta})$. Let $\pi_k$ denote the policy induced by parameters $\boldsymbol{\theta}_k$, and $d_k$ denote the corresponding discounted state-action distribution. The vanilla policy gradient is given by (Sutton et al., 1999):

$$\nabla_{\boldsymbol{\theta}}^{\text{PG}} J(\boldsymbol{\theta}) := \mathbb{E}_{(s,a) \sim d_\pi} \left[ \frac{\partial}{\partial \boldsymbol{\theta}} \log \pi(a|s; \boldsymbol{\theta}) A_\pi(s, a) \right].$$

When the parameter space has a non-Euclidean geometry, the vanilla gradient does not correspond to the direction of steepest ascent. To address this, Amari (1998) proposed the natural gradient, which accounts for the information geometry of the parameter manifold. In reinforcement learning, Kakade (2001) introduced the natural policy gradient based on the Fisher matrix

$$\boldsymbol{F}(\boldsymbol{\theta}) := \mathbb{E}_{d_\pi(s,a)} \left[ \frac{\partial}{\partial \boldsymbol{\theta}} \log \pi(a|s; \boldsymbol{\theta}) \frac{\partial}{\partial \boldsymbol{\theta}^\top} \log \pi(a|s; \boldsymbol{\theta}) \right].$$

The resulting natural policy gradient is defined as

$$\nabla_{\boldsymbol{\theta}}^{\text{NPG}} J(\boldsymbol{\theta}) := \boldsymbol{F}(\boldsymbol{\theta})^{-1} \nabla_{\boldsymbol{\theta}}^{\text{PG}} J(\boldsymbol{\theta}).$$

We focus on Kakade's formulation of the natural policy gradient, which has been shown to be invariant to reparameterization (Bagnell & Schneider, 2003). Following Kunstner et al. (2019), we refer to the Fisher matrix estimated from finite samples as the *empirical Fisher*, and call the resulting gradients the *Empirical Natural Policy Gradients*.

**Randomized block Kaczmarz method.**  The randomized block Kaczmarz method is an iterative algorithm for solving overdetermined least-squares problems of the form:

$$\min_{\boldsymbol{x}} \|\boldsymbol{A}\boldsymbol{x} - \boldsymbol{b}\|_2^2, \tag{1}$$

where $\boldsymbol{A} \in \mathbb{R}^{n \times d}$ and $\boldsymbol{b} \in \mathbb{R}^n$. Starting from an initial estimate $\boldsymbol{x}_0$, the method iteratively refines the solution by projecting onto the solution space of randomly selected row blocks of $\boldsymbol{A}$. Specifically, let $T = \{\tau_1, \tau_2, \ldots, \tau_m\}$ be a partition of the rows of $\boldsymbol{A}$. At iteration $j$, a block $\tau_j \in T$ is sampled (typically uniformly at random), and the update is:

$$\boldsymbol{x}_j = \boldsymbol{x}_{j-1} + \boldsymbol{A}_{\tau_j}^\top (\boldsymbol{A}_{\tau_j} \boldsymbol{A}_{\tau_j}^\top)^{-1} (\boldsymbol{b}_\tau - \boldsymbol{A}_{\tau_j} \boldsymbol{x}_{j-1}).$$

Under standard assumptions, this randomized block scheme converges to the least-squares solution, often with favorable convergence properties compared to deterministic variants (Needell & Tropp, 2014).

## 4. Randomized Advantage Transformation

In this section, we first show that Tikhonov-regularized natural policy gradients (NPG) can be formulated as a regularized least-squares problem. We then apply the Woodbury formula to express the resulting NPG update in terms of a transformed advantage, reducing it to a vanilla policy gradient form. Finally, we introduce a randomized Kaczmarz iteration to efficiently solve the corresponding least squares.

### 4.1. Tikhonov Regularized NPG

As shown by Kakade (2001), the natural policy gradient can be obtained as the solution to a least squares problem. We follow the notation of Schulman et al. (2017a). Use $n$ to denote the cardinality of state-action space, i.e., $n = |\mathcal{S}||\mathcal{A}|$, and $p$ to denote the number of policy parameters, i.e., $p = |\boldsymbol{\theta}|$. Let $\boldsymbol{\Sigma} \in \mathbb{R}^{n \times n}$ be a diagonal matrix with diagonal entries $d_\pi(s)\pi(a|s)$, $\boldsymbol{H} \in \mathbb{R}^{n \times p}$ be the matrix whose rows are $\frac{\partial}{\partial \boldsymbol{\theta}^\top} \log \pi(a|s; \boldsymbol{\theta})$, $\boldsymbol{y} \in \mathbb{R}^n$ be the vector with entries $A_\pi(s, a)$. Under this notation, the vanilla PG and NPG can be written as:

$$\text{PG: } \nabla_{\boldsymbol{\theta}}^{\text{PG}} J(\theta) \coloneqq \boldsymbol{H}^\top \boldsymbol{\Sigma} \boldsymbol{y}, \tag{2}$$

$$\text{NPG: } \nabla_{\boldsymbol{\theta}}^{\text{NPG}} J(\theta) \coloneqq (\boldsymbol{H}^\top \boldsymbol{\Sigma} \boldsymbol{H})^{-1} \boldsymbol{H}^\top \boldsymbol{\Sigma} \boldsymbol{y}. \tag{3}$$

The NPG update coincides with the solution to the following least squares.

**Proposition 1** (Kakade (2001)). *The minimizer of* $\min_{\boldsymbol{x}} \|\boldsymbol{y} - \boldsymbol{H}\boldsymbol{x}\|_{\boldsymbol{\Sigma}}^2$ *is given by* $x^* = (\boldsymbol{H}^\top \boldsymbol{\Sigma} \boldsymbol{H})^{-1} \boldsymbol{H}^\top \boldsymbol{\Sigma} \boldsymbol{y}$, *provided the inverse exists.*

In practice, the Fisher matrix is estimated from a finite number of samples, (i.e., empirical Fisher), which can render it ill-conditioned or singular. To stabilize inversion, it is common to apply Tikhonov regularization by adding a damping term (Schulman et al., 2015; Martens & Grosse, 2015).

$$\text{T-NPG: } \nabla_{\boldsymbol{\theta}}^{\text{T-NPG}} J(\theta) \coloneqq (\lambda \boldsymbol{I} + \boldsymbol{H}^\top \boldsymbol{\Sigma} \boldsymbol{H})^{-1} \boldsymbol{H}^\top \boldsymbol{\Sigma} \boldsymbol{y}, \tag{4}$$

where $\lambda > 0$ is a damping coefficient. This Tikhonov-regularized NPG (T-NPG) is equivalently the solution to the regularized least-squares:

$$(\lambda \boldsymbol{I} + \boldsymbol{H}^\top \boldsymbol{\Sigma} \boldsymbol{H})^{-1} \boldsymbol{H}^\top \boldsymbol{\Sigma} \boldsymbol{y}$$
$$= \arg\min_{\boldsymbol{g}} \|\boldsymbol{y} - \boldsymbol{H}\boldsymbol{g}\|_{\boldsymbol{\Sigma}}^2 + \lambda \|\boldsymbol{g}\|_2^2 \tag{5}$$

This formulation motivates our use of iterative least-squares solvers in the following section.

### 4.2. Woodbury Reformulation

Interestingly, the introduction of Tikhonov regularization enables the use of the Woodbury formula to transform the inverse (Wu et al., 2024; Guzmán-Cordero et al., 2025). Applying the formula, $(\boldsymbol{I} + \boldsymbol{U}\boldsymbol{V})^{-1}\boldsymbol{U} = \boldsymbol{U}(\boldsymbol{I} + \boldsymbol{V}\boldsymbol{U})^{-1}$ for any conformable matrices $\boldsymbol{U}, \boldsymbol{V}$, twice yields

$$\nabla_{\boldsymbol{\theta}}^{\text{T-NPG}} J(\boldsymbol{\theta}) = (\lambda \boldsymbol{I}_p + \boldsymbol{H}^\top \boldsymbol{\Sigma} \boldsymbol{H})^{-1} \boldsymbol{H}^\top \boldsymbol{\Sigma} \boldsymbol{y} \tag{6}$$

$$= \boldsymbol{H}^\top (\lambda \boldsymbol{I}_n + \boldsymbol{\Sigma} \boldsymbol{H} \boldsymbol{H}^\top)^{-1} \boldsymbol{\Sigma} \boldsymbol{y} \tag{7}$$

$$= \boldsymbol{H}^\top \boldsymbol{\Sigma} \boxed{(\lambda \boldsymbol{I}_n + \boldsymbol{H} \boldsymbol{H}^\top \boldsymbol{\Sigma})^{-1} \boldsymbol{y}}. \tag{8}$$

Compared to vanilla policy gradients $\nabla_{\boldsymbol{\theta}}^{\text{PG}} J(\boldsymbol{\theta}) = \boldsymbol{H}^\top \boldsymbol{\Sigma} \boldsymbol{y}$, the only difference is the advantage term. Specifically, T-NPG can be written as $\nabla_{\boldsymbol{\theta}}^{\text{T-NPG}} J(\theta) = \boldsymbol{H}^\top \boldsymbol{\Sigma} \tilde{\boldsymbol{y}}$ where

$$\tilde{\boldsymbol{y}} = (\lambda \boldsymbol{I}_n + \boldsymbol{H} \boldsymbol{H}^\top \boldsymbol{\Sigma})^{-1} \boldsymbol{y}. \tag{9}$$

Thus, the T-NPG corresponds to a vanilla policy gradient with a *transformed advantage function*,

$$\bar{A}_\pi(s, a) \coloneqq \left[ (\lambda \boldsymbol{I}_n + \boldsymbol{H} \boldsymbol{H}^\top \boldsymbol{\Sigma})^{-1} \boldsymbol{y} \right]_{(s,a)}. \tag{10}$$

Importantly, the matrix inversion in the above transformation does not depend on the number of parameters $p$ since $\lambda \boldsymbol{I}_n + \boldsymbol{H} \boldsymbol{H}^\top \boldsymbol{\Sigma}$ is of size $n \times n$. This important property is the key to develop our method.

### 4.3. Randomized Kaczmarz Iteration

A major limitation of the above advantage transformation is that $n$ is typically much larger than $p$, essentially in continuous state-action spaces, making the exact transformation infeasible. Since T-NPG naturally arises from a least-squares problem, we turn to randomized iterative solvers for linear systems (Gower & Richtárik, 2015) and propose to approximate the advantage transformation via randomization.

Equation (5) involves weighting by $\boldsymbol{\Sigma}$, which is generally unknown. To bypass this, we replace this objective with an unweighted least squares problem constructed from on-policy samples. This corresponds to a Monte Carlo approximation in which state-action pairs are drawn i.i.d from $d_\pi(s)\pi(a|s)$, so that expectations over $\boldsymbol{\Sigma}$ are replaced by empirical averages. Specifically, sampling $(s, a) \sim d_\pi(s)\pi(a|s)$ yields

$$\mathbb{E}_\tau[\|\boldsymbol{y}_\tau - \boldsymbol{H}_\tau \boldsymbol{g}\|_2^2] \propto \|\boldsymbol{y} - \boldsymbol{H}\boldsymbol{g}\|_{\boldsymbol{\Sigma}}^2. \tag{11}$$

Under standard on-policy assumptions, the resulting estimator is unbiased in expectation, and the discrepancy introduced by ignoring $\boldsymbol{\Sigma}$ vanishes as the batch size increases.

To solve Equation (5), we adopt randomized block Kaczmarz method (Needell & Tropp, 2014). Let $\mathcal{D}_k$ denote on-policy samples at iteration $k$, partitioned into mini-batches $\{\tau_1, \tau_2, \ldots, \tau_m\}$. Starting from an initial estimate $\boldsymbol{g}_0$, at iteration $j$, we select a batch $\tau_j$, and perform the following:

$$\boldsymbol{g}_j \leftarrow \arg\min_{\boldsymbol{g}} \|\boldsymbol{y}_{\tau_j} - \boldsymbol{H}_{\tau_j} \boldsymbol{g}\|_2^2 + \lambda \|\boldsymbol{g} - \boldsymbol{g}_{j-1}\|_2^2. \tag{12}$$

Here, $\boldsymbol{H}_{\tau_j}$ denotes the rows of $\boldsymbol{H}$ indexed by $\tau_j$. Note that this corresponds to a regularized block update in the randomized block Kaczmarz framework. In the classical (unregularized) formulation, each step projects the current iterate onto the solution space of the sampled linear system $\boldsymbol{H}_{\tau_j} \boldsymbol{g} = \boldsymbol{y}_{\tau_j}$. However, enforcing this hard constraint directly can be unstable in our setting due to noise and potential rank deficiency of minibatches. Thus, we consider the regularized version, which can be viewed as a proximal update that balances fitting the current batch with staying close to the previous estimate. Importantly, as shown in Goldshlager et al. (2024), Equation (12) admits a closed-form

update

$$\boldsymbol{g}_j \leftarrow \boldsymbol{g}_{j-1} + \boldsymbol{H}_{\tau_j}^\top \underbrace{\left( \lambda \boldsymbol{I} + \boldsymbol{H}_{\tau_j} \boldsymbol{H}_{\tau_j}^\top \right)^{-1} \left( \boldsymbol{y}_{\tau_j} - \boldsymbol{H}_{\tau_j} \boldsymbol{g}_{j-1} \right)}_{\text{Randomized Advantage Transformation}}.$$
(13)

The bracketed term performs an advantage transformation on the sampled data. We therefore refer to this method as *Randomized Advantage Transformation* (RAT), i.e.,

$$\tilde{A}_j(s,a) \coloneqq \left[ \left( \lambda \boldsymbol{I} + \boldsymbol{H}_{\tau_j} \boldsymbol{H}_{\tau_j}^\top \right)^{-1} \left( \boldsymbol{y}_{\tau_j} - \boldsymbol{H}_{\tau_j} \boldsymbol{g}_{j-1} \right) \right]_{(s,a)}.$$
(14)

With minibatch size $B$ we have $\boldsymbol{H}_\tau \in \mathbb{R}^{B \times p}$ and $\boldsymbol{y}_\tau \in \mathbb{R}^B$. When $B \ll p$, the matrix inversion is only $B \times B$, in contrast to the original $n \times n$ matrix. The main computational cost arises from forming $\boldsymbol{H}_\tau \boldsymbol{H}_\tau^\top$, which scales as $\mathcal{O}(pB^2)$; this cost can be further reduced using Nyström Approximation (Gittens & Mahoney, 2013). Further, as pointed by Guzmán-Cordero et al. (2025), $\boldsymbol{H}\boldsymbol{H}^\top$ is the neural tangent kernel (Jacot et al., 2018), and thus can be approximated efficiently in various ways (Novak et al., 2022). $\boldsymbol{H}$ can be estimated efficiently using the per-sample gradients in PyTorch[1]. The advantage can be computed via `torch.linalg.solve`: which directly solve linear system with matrix $(\lambda \boldsymbol{I} + \boldsymbol{H}\boldsymbol{H}^\top)$ and vector $\boldsymbol{y}$. It is faster and more numerically stable than explicitly computing the inverse.

The natural policy gradients can then be computed via direct backpropagation using the following PPO-like objective:

$$J_{\text{RAT}}(\boldsymbol{\theta}) \coloneqq \mathbb{E}_{(s,a) \sim \mathcal{D}_k} \left[ \frac{\pi(a|s; \boldsymbol{\theta})}{\pi_{\text{old}}(a|s)} \tilde{A}_j(s,a) \right],$$
(15)

where $\pi_{\text{old}}$ is the behavior policy used to collect $\mathcal{D}_k$, and remains the same during the inner iterations.

> **Intuition Underlying RAT.** RAT can be viewed as an efficient method for constructing a compatible approximation of advantage function by solving linear system $\boldsymbol{H}\boldsymbol{x} = \boldsymbol{y}$ with Tikhonov regularization. At each iteration, RAT updates the current estimate $\boldsymbol{g}_{j-1}$ by projecting the residual $\boldsymbol{y}_\tau - \boldsymbol{H}_\tau \boldsymbol{g}_{j-1}$ onto the solution space of $\boldsymbol{H}_\tau \boldsymbol{x} = \boldsymbol{y}_\tau$, with Tikhonov regularization. This projection implicitly injects curvature information into the advantage estimates. By iterating over mini-batches, RAT progressively aggregates local curvature information, yielding an accurate approximation of the full natural policy gradient without explicitly forming or inverting the Fisher matrix.

Importantly, RAT differs from SPRING (Goldshlager et al., 2024) in two key aspects. First, RAT has no momentum in-

---

[1] https://docs.pytorch.org/tutorials/intermediate/per_sample_grads.html

terpretation, whereas SPRING can be viewed as a Kaczmarz-inspired momentum-based method. Second, RAT performs inner iterations over mini-batches within a single on-policy rollout, whereas SPRING applies a single update per batch. This distinction is crucial as it enables iterative refinement of the NPG estimate without cumulating $\boldsymbol{g}_j$ across rollouts.

As in KFAC (Martens & Grosse, 2015), gradient norm clipping is essential for stability. Instead of clipping in the Fisher norm, we clip $\ell_2$-norm of the estimated gradients $\boldsymbol{g}_j$ (Zhang et al., 2020): $\alpha_j \coloneqq \min \left( \eta, \frac{\nu}{\|\boldsymbol{g}_j\|_2} \right)$, where $\nu > 0$ is a threshold and $\eta > 0$ is the learning rate.

**Shared actor-critic architectures.** We follow Wu et al. (2017) and estimate joint natural policy gradients by modeling the value function as a Gaussian. To apply RAT to the critic, we explicitly introduce a pseudo advantage (e.g., an all-ones vector). RAT is then applied jointly to the policy advantage and the critic pseudo advantage, yielding a unified and stable optimization objective for shared actor-critic networks. We refer readers to Appendix C.1 for details. This joint application of RAT to both policy and critic updates clearly shows the flexibility of the advantage transformation, and distinguishes our method from prior work on Woodbury-based approaches (Guzmán-Cordero et al., 2025). Specifically, while Guzmán-Cordero et al. (2025) can in principle be applied in this setting, they typically require maintaining separate curvature-adjusted gradients for actor and critic and carefully merging them during parameter updates. This merging is inherently architecture-dependent, as it requires explicit knowledge of which parameters are shared and how gradients from different heads should be combined. In contrast, RAT introduces a pseudo-advantage formulation that unifies the actor and critic objectives into a single surrogate loss. The resulting gradient is computed via standard backpropagation. As a result, curvature-adjusted updates are handled implicitly by `autograd`, without requiring manual gradient partitioning or architecture-specific merging logic. This allows RAT to remain architecture-agnostic in practice, even in shared-network settings. The full RAT is summarized in Algorithm 1 in Appendix.

### 4.4. Convergence Results of RAT

RAT is closely related to randomized block Kaczmarz method (Needell & Tropp, 2014) and, more generally, to the class of randomized iterative methods (Gower & Richtárik, 2015). Its convergence analysis can be viewed as an extension of these methods to the setting of Tikhonov-regularized natural policy gradients.

We begin with a standard assumption on the matrix $\boldsymbol{H}$.

**Assumption 1** (Full column rank). *The full data matrix $\boldsymbol{H} \in \mathbb{R}^{n \times p}$ satisfies $\text{rank}(\boldsymbol{H}) = p$ and $p \ll n$.*

This assumption is consistent with the common setting in reinforcement learning in which the state-action space is large or continuous, and function approximation is needed.

**Assumption 2** (State-action coverage). *Let $\boldsymbol{h}_i^\top$ denote $i$-th row of $\boldsymbol{H}$, and $\tau$ denote a random minibatch sampled from $d_\pi(s,a)$. For each index $i \in \{1, \ldots, n\}$, $\mathbb{P}(i \in \tau) > 0$.*

This assumption ensures that every state-action pair has a non-zero probability of being sampled, which is a standard requirement for defining the natural policy gradient (Bagnell & Schneider, 2003; Kakade, 2001).

Under the above assumptions, we obtain the following:

**Lemma 1.** *Define $\boldsymbol{P}_\tau := \boldsymbol{H}_\tau^\top(\lambda\boldsymbol{I} + \boldsymbol{H}_\tau\boldsymbol{H}_\tau^\top)^{-1}\boldsymbol{H}_\tau$, then*

$$\mu := \lambda_{\min}(\mathbb{E}[\boldsymbol{P}_\tau]) > 0.$$

We now present two theorems characterizing the convergence behavior of RAT. We first analyze an idealized case in which the advantage is *exactly compatible* (Peters & Schaal, 2008). Specifically, for all minibatches $\tau$, $\boldsymbol{y}_\tau = \boldsymbol{H}_\tau\boldsymbol{g}^*$, where $\boldsymbol{g}^*$ is the solution to Equation (5).

**Theorem 1** (Linear convergence of RAT). *Assume minibatches $\tau_j$ are sampled i.i.d. from $d_\pi(s,a)$. Then*

$$\mathbb{E}\|\boldsymbol{g}_j - \boldsymbol{g}^*\|_2^2 \le (1 - \mu)^j\|\boldsymbol{g}_0 - \boldsymbol{g}^*\|_2^2. \qquad (16)$$

This theorem, together with Lemma 1, guarantees that the RAT update is contractive, which is essential for establishing linear convergence. The convergence rate of RAT is entirely characterized by the spectrum of $\mathbb{E}[\boldsymbol{P}_\tau]$. In particular, as $\boldsymbol{P}_\tau = \boldsymbol{H}_\tau^\top\boldsymbol{H}_\tau(\lambda\boldsymbol{I} + \boldsymbol{H}_\tau^\top\boldsymbol{H}_\tau)^{-1}$, when $\boldsymbol{H}_\tau^\top\boldsymbol{H}_\tau$ (empirical Fisher) has a low rank, a smaller $\lambda$ generally leads to a larger $\mu$, and thus faster convergence. This behavior is also observed in our sensitivity analysis. In addition, Assumption 1 applies to the *full matrix $\boldsymbol{H}$*, not to individual minibatches. In practice, minibatch matrices $\boldsymbol{H}_\tau$ can be low-rank. The Tikhonov damping term $\lambda > 0$ ensures that $(\lambda\boldsymbol{I} + \boldsymbol{H}_\tau\boldsymbol{H}_\tau^\top)$ is always invertible, even when $\boldsymbol{H}_\tau$ is rank-deficient. Poor state-action coverage affects the convergence rate through $\mu = \lambda_{\min}(\mathbb{E}[\boldsymbol{P}_\tau])$, but does not invalidate the analysis.

We now consider the more realistic case in which the advantage estimates are noisy: $\boldsymbol{y}_\tau = \boldsymbol{H}_\tau\boldsymbol{g}^* + \boldsymbol{\xi}_\tau$, where $\boldsymbol{\xi}_\tau$ is a zero-mean random variable satisfying $\mathbb{E}[\boldsymbol{\xi}_\tau \mid \tau] = \boldsymbol{0}$.

**Theorem 2** (Convergence with error floor). *Define $\eta^2 := \mathbb{E}\big[\|\boldsymbol{H}_\tau^\top(\lambda\boldsymbol{I} + \boldsymbol{H}_\tau\boldsymbol{H}_\tau^\top)^{-1}\boldsymbol{\xi}_\tau\|_2^2\big]$. Then*

$$\mathbb{E}\|\boldsymbol{g}_j - \boldsymbol{g}^*\|_2^2 \le (1 - \mu)^j\|\boldsymbol{g}_0 - \boldsymbol{g}^*\|_2^2 + \frac{\eta^2}{\mu}. \qquad (17)$$

$\eta^2$ effectively quantifies the norm discrepancy between the true gradient and its stochastic estimate. This motivates the use of gradient norm clipping in practice.

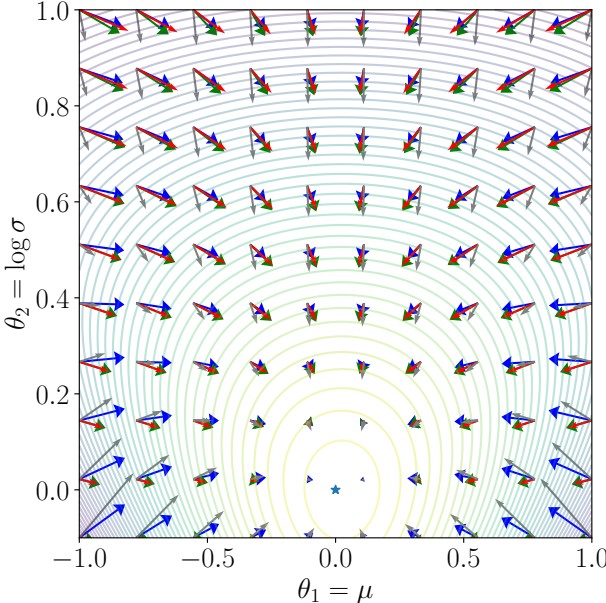

*Figure 2.* Univariate Gaussian with $\theta_1 = \mu$ and $\theta_2 = \log\sigma$ (closed-formed natural gradients, empirical natural gradients, RAT gradients and vanilla gradients; $\star$ for the optimum). RAT closely approximates empirical natural gradients.

It is worth noting that Algorithm 1 in Appendix implements multiple inner iterations per batch, resulting in a time-varying sequence of linear systems. We show in Appendix C.2 that the practical implementation can be interpreted as a *contractive solver tracking a slowly varying sequence of systems*. Let $\boldsymbol{g}_t^*$ denote the solution of the regularized least-squares problem defined by the current policy $\boldsymbol{\theta}_t$, and define the tracking error $e_t := \|\boldsymbol{g}_t - \boldsymbol{g}_t^*\|$. Under our settings (small learning rates and gradient clipping), the tracking error remains small, yielding a bounded steady-state error of order $\mathcal{O}(\max_t\|\boldsymbol{\theta}_{t+1} - \boldsymbol{\theta}_t\|)$. This aligns with standard analyses of stochastic approximation in RL, where updates track a moving target induced by policy changes, and provides a heuristic justification for the implementation.

# 5. Experiments

We evaluate Randomized Advantage Transformation (RAT) through a combination of controlled illustrations, continuous control benchmarks, and high-dimensional visual domains. Our goals are to: (1) verify that RAT accurately approximates empirical natural gradients, (2) assess its empirical performance and efficiency relative to established natural policy gradients methods, and (3) analyze the sensitivity to its key design choices.

Across all experiments, we apply standard stabilization techniques, including observation normalization (Mnih et al., 2016), advantage normalization (Schulman et al., 2017b),

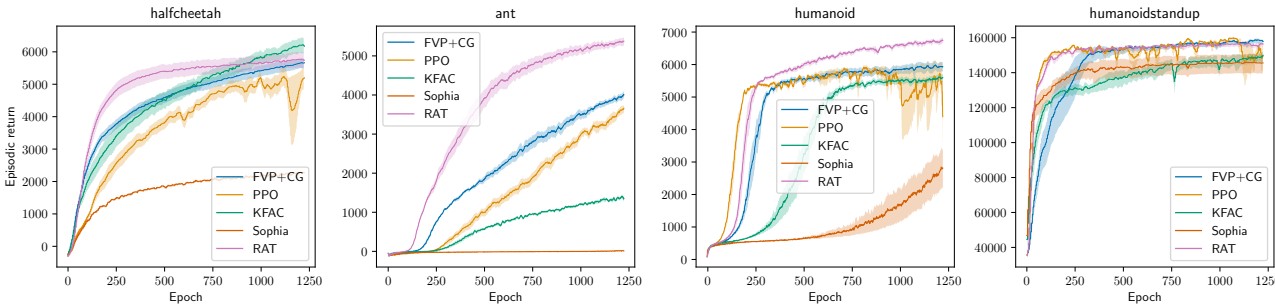

*Figure 3.* Optimizing MLP policies on continuous control tasks with separate actor-critic networks. RAT outperforms KFAC, FVP+CG and Sophia in most tasks. The shaded region denotes the standard error over 5 random seeds.

*Table 1.* Final performance (mean ± stderr over 5 seeds) of different methods on continuous controls with shared actor-critic networks.

| Ep. Returns | Swimmer ↑ | Hopper ↑ | HalfCheetah ↑ | Walker2d ↑ | Ant ↑ | Humanoid ↑ | HumanoidStandup ↑ |
|---|---|---|---|---|---|---|---|
| $\mathcal{S} \times \mathcal{A}$ | $8 \times 2$ | $11 \times 3$ | $17 \times 6$ | $17 \times 6$ | $105 \times 8$ | $376 \times 17$ | $376 \times 17$ |
| **RAT (Ours)** | $271.6^{\pm 36.3}$ | $2334.6^{\pm 524.9}$ | $4629.2^{\pm 287.4}$ | $3156.0^{\pm 293.6}$ | $2926.6^{\pm 353.1}$ | $5382.7^{\pm 117.3}$ | $146529.7^{\pm 2317.6}$ |
| **ACKTR** | $59.1^{\pm 13.0}$ | $2138.9^{\pm 171.6}$ | $3630.9^{\pm 282.6}$ | $2576.6^{\pm 154.6}$ | $23.4^{\pm 3.2}$ | $2571.7^{\pm 838.7}$ | $127928.5^{\pm 5433.7}$ |
| **PPO** | $191.3^{\pm 32.7}$ | $2346.8^{\pm 202.7}$ | $4146.0^{\pm 107.5}$ | $2225.3^{\pm 303.4}$ | $1373.9^{\pm 26.0}$ | $5357.9^{\pm 150.9}$ | $130014.2^{\pm 6463.7}$ |
| **Sophia** | $57.9^{\pm 5.9}$ | $1104.0^{\pm 90.6}$ | $899.5^{\pm 113.2}$ | $1256.0^{\pm 129.7}$ | $-7.0^{\pm 1.4}$ | $669.4^{\pm 56.2}$ | $111212.6^{\pm 13449.9}$ |

and PopArt for value normalization (Hessel et al., 2019). Unless stated otherwise, all methods use the same network architectures and training pipelines. We report results averaged over five random seeds for a fixed training budget of 1250 epochs, which corresponds to approximately 10 million environment steps (a standard budget in continuous control benchmarks). Additional implementation details are provided in Appendix C. The code for reproducing our results is available at Code URL[2].

### 5.1. Illustration of Natural Gradients Estimation

We begin with a low-dimensional example that admits an analytic form of the natural gradient, enabling direct visualization of the update directions. Specifically, we consider maximum-likelihood estimation for a univariate Gaussian parameterized by its mean and log-standard deviation: $\mathcal{N}(x|\mu, \sigma; \theta_1, \theta_2)$, where $\mu = \theta_1$ and log standard deviation $\log \sigma = \theta_2$. In this setting, the inverse Fisher matrix is available in closed form, allowing us to compute exact natural gradients (see Appendix A for details). Figure 2 compares the vanilla gradients, the natural gradient, the empirical natural gradients and the gradient estimated by RAT. We also plot the contour lines for $\log \mathcal{N}(x|\theta)$ (loss landscape) to better illustrate how the natural gradient differs from the vanilla gradient. While vanilla gradients follow the steepest ascent direction of $\log \mathcal{N}(x|\theta)$, perpendicular to the contour lines, the natural gradient accounts for the geometry induced by the parameterization and points more directly towards the

optimum. The updates produced by RAT closely match the empirical natural gradients, demonstrating the effectiveness and accuracy of RAT with finite samples.

### 5.2. Continuous Control with MLP Policies

We next evaluate RAT on standard continuous control benchmarks from OpenAI Gym (Brockman et al., 2016) implemented in MuJoCo (Todorov et al., 2012). We consider Walker2d-v4 ($\mathcal{A} \in \mathbb{R}^6$), HalfCheetah-v4 ($\mathcal{A} \in \mathbb{R}^6$), Ant-v4 ($\mathcal{A} \in \mathbb{R}^8$), and Humanoid-v4 ($\mathcal{A} \in \mathbb{R}^{17}$), which span action dimensions from 6 to 17. Policies and value functions are parameterized by two-layer MLPs with 256 hidden units and Tanh activations; the policy outputs the mean of a Gaussian distribution, with a state-independent log standard deviation. We compare RAT against several strong baselines for estimating natural policy gradients, including Fisher-vector products with conjugate gradient (FVP+CG) from (Schulman et al., 2015), Kronecker-Factored Approximate Curvature (KFAC) (Martens & Grosse, 2015), and a diagonal Fisher approximation (i.e., $\text{diag}(\lambda \mathbf{I} + \mathbf{H}^\top \mathbf{H})$) according to Sophia (Liu et al., 2024). All methods are implemented within the same codebase, and baseline hyperparameters are tuned for best performance.

**Separate actor-critic networks.** Figure 3 reports learning curves when actor and critic are optimized separately. RAT consistently matches or outperforms all baselines across all tasks, exhibiting both faster learning and higher final returns. In particular, RAT remains stable on challenging tasks such as Ant-v4 and Humanoid-v4, whereas KFAC frequently

---

[2]https://github.com/agent-lab/ICML2026-RAT

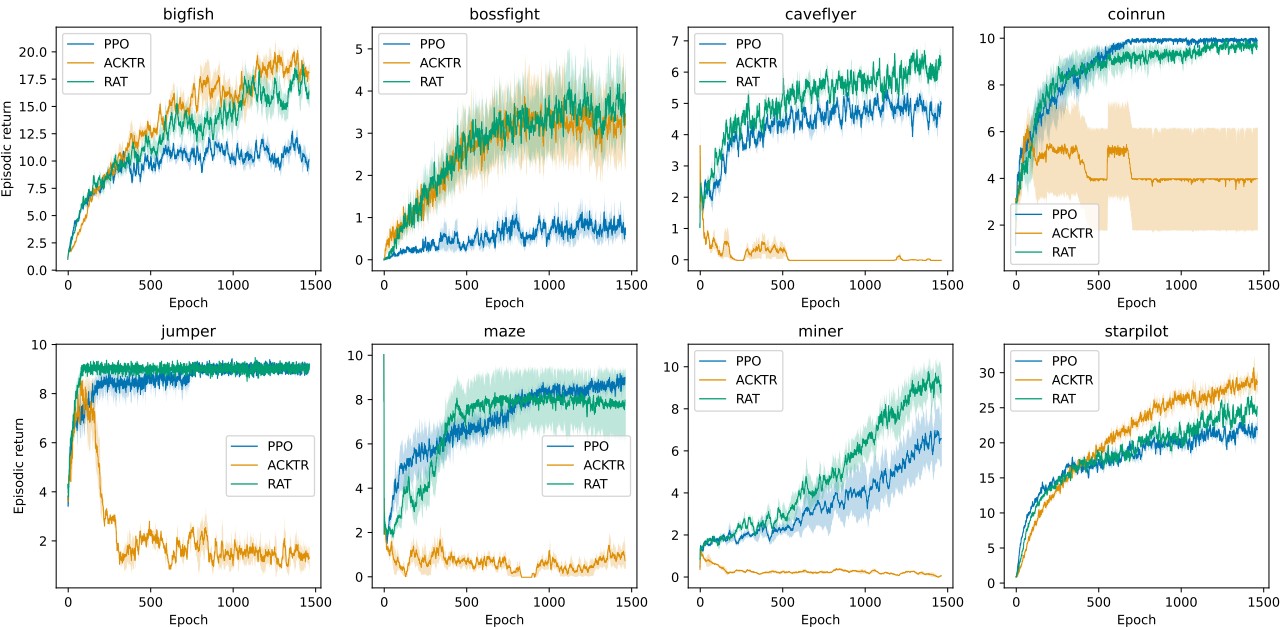

*Figure 4.* Optimizing ResNet policies for discrete controls in ProcGen environments: RAT performs consistently well across all 8 tasks, delivering comparable or higher episodic returns than all baselines. The shaded region denotes the standard error over 5 random seeds.

*Table 2.* Wallclock time per update (in ms) on continuous control tasks with separate actor-critic networks.

| | Time (ms) | HalfCheetah ↓ | Ant ↓ | Humanoid ↓ |
|---|---|---|---|---|
| **Separate** | **RAT (Ours)** | $9.83^{\pm1.49}$ | $10.04^{\pm1.35}$ | $18.17^{\pm3.04}$ |
| | **FVP+CG** | $19.86^{\pm1.15}$ | $19.95^{\pm1.20}$ | $19.81^{\pm1.18}$ |
| | **KFAC** | $5.60^{\pm1.28}$ | $5.61^{\pm1.23}$ | $6.57^{\pm1.47}$ |
| | **Sophia** | $3.92^{\pm0.71}$ | $3.98^{\pm0.73}$ | $5.71^{\pm0.68}$ |
| | **PPO** | $3.12^{\pm1.38}$ | $3.18^{\pm1.36}$ | $3.22^{\pm1.40}$ |
| **Shared** | **RAT (Ours)** | $11.53^{\pm1.69}$ | $11.66^{\pm1.55}$ | $19.85^{\pm3.11}$ |
| | **ACKTR** | $6.92^{\pm1.63}$ | $6.85^{\pm1.55}$ | $7.87^{\pm1.70}$ |
| | **Sophia** | $5.97^{\pm0.94}$ | $6.03^{\pm1.02}$ | $7.58^{\pm0.97}$ |
| | **PPO** | $3.70^{\pm1.56}$ | $3.70^{\pm1.46}$ | $3.72^{\pm1.49}$ |

learns slowly. We also evaluated an enhanced variant of KFAC, i.e., eKFAC (George et al., 2018), and found that the eKFAC method did not yield noticeable improvements over KFAC in our experiments, see Figure 6 in Appendix E. The Sophia method performs poorly on all tasks, highlighting the importance of capturing parameter correlations. In addition, RAT performs significantly better than PPO on challenging Ant and Humanoid tasks.

**Shared actor-critic networks.** We further evaluate RAT in the shared-network setting, where curvature estimation is more challenging. Since FVP+CG is not directly applicable, we compare against Proximal Policy Optimization (PPO) (Schulman et al., 2017b), a simplified approximation to natural policy gradients (Hilton et al., 2022), ACKTR (Wu

et al., 2017), an extended KFAC method for shared networks, and Sophia (Liu et al., 2024). Table 1 summarizes final performance. RAT achieves the best overall returns on most tasks, with substantial gains on Ant and Humanoid, highlighting its robustness in challenging shared architectures.

Table 2 reports wallclock time per update (in ms; averaged over 124 updates on Xeon(R) w5-2445 GeForce RTX 4090). While the PPO is the fastest, RAT is significantly more efficient than FVP+CG and offers a favorable trade-off between computational cost and performance. Despite higher per-update cost, RAT is most beneficial in regimes where curvature matters (e.g., high-dimensional settings), where PPO often plateaus or requires careful tuning. RAT is not designed to match PPO's per-step efficiency, but to provide a simple, architecture-agnostic, and principled approximation to natural policy gradients. Compared to existing natural-gradient methods, it offers a stronger performance–compute trade-off while avoiding architecture-specific approximations and complex inner solvers.

## 5.3. Visual Control with ResNet Policies

To assess scalability to high-dimensional visual inputs, we evaluate RAT on the challenging Procgen Benchmark (Cobbe et al., 2020), which features procedurally generated environments with 64x64 RGB observations. We consider 8 representative environments, including *BigFish*, *BossFight*, *CaveFlyer*, *Climber*, *Dodgeball*, *FruitBot*, *Heist*, and *StarPilot*. Policies are parameterized using a ResNet-

based architecture, adapted from Espeholt et al. (2018), and training follows the standard Procgen protocol for evaluating sample efficiency, i.e., training and testing on the same distribution of levels in each environment.

Figure 4 presents learning curves comparing RAT with PPO and ACKTR. RAT performs consistently well across all tasks, delivering comparable or higher returns than all baselines while avoiding training instabilities frequently observed in KFAC-based methods. These results demonstrate that RAT scales effectively to visual and complex dynamics.

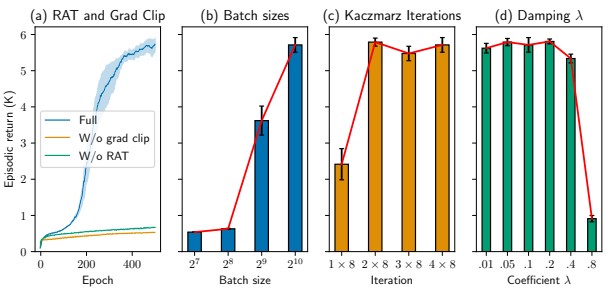

*Figure 5.* Ablation and sensitivity analysis of RAT on Humanoid.

### 5.4. Ablations and Sensitivity Analysis

Finally, we conduct an ablation study and sensitivity analysis to pinpoint the influence of key components and hyperparameters of RAT on performance, focusing on the challenging Humanoid task. Figure 5(a) presents ablations that remove either the advantage transformation or gradient norm clipping. Both components are essential: removing either leads to substantial performance degradation. We further study sensitivity of RAT to batch size, number of Kaczmarz iterations and damping coefficient $\lambda$. The results are presented in Figure 5(b), (c) and (d). RAT benefits from sufficiently large batch sizes ($2^{10}$) (as the batch size effectively determines the rank of empirical Fisher matrix), while remaining relatively robust to the number of Kaczmarz iterations within a reasonable range (too few iterations result in suboptimal performance). The performance of RAT is also robust across a broad range of damping values (from 0.01 to 0.4), with degradation only occurring at very large values (e.g., $\lambda = 0.8$). More analysis on Ant can be found in Figure 8 in Appendix. Overall, these results indicate that RAT is robust and does not require fine-grained tuning.

## 6. Limitations and Conclusion

**Limitations.** Despite strong theoretical and empirical results, RAT has several limitations. First, RAT requires forming minibatch-level matrices $\boldsymbol{H}_\tau \boldsymbol{H}_\tau^\top$, with cost $\mathcal{O}(pB^2)$. While substantially cheaper than full Fisher inversion and architecture-agnostic, this can become a bottleneck for large policies or batch sizes; low-rank or sketch-based approxi-

mations may improve scalability. Second, the convergence rate of RAT depends on the minimum singular value of $\boldsymbol{P}_\tau$. Poorly conditioned minibatches may slow convergence, although gradient norm clipping alleviates this issue in practice. Third, our analysis focus on the on-policy setting; extending RAT to full off-policy settings where the advantage function is estimated from replay buffer samples remains an open direction.

**Conclusion.** We proposed Randomized Advantage Transformation (RAT), an efficient and architecture-agnostic method for estimating natural policy gradients. By using a Woodbury-based reformulation, RAT transforms curvature information into the advantage function and estimates regularized natural policy gradients using standard backpropagation, without explicit Fisher construction or conjugate-gradient solvers. We provided convergence guarantees and demonstrated strong empirical performance on continuous control and high-dimensional visual benchmarks. RAT bridges the gap between principled natural gradient methods and practical deep reinforcement learning, offering a simple and scalable alternative for second-order policy optimization.

## Impact Statement

This work advances scalable optimization methods for reinforcement learning by enabling efficient computation of natural policy gradients without explicit curvature estimation. By simplifying implementation and reducing computational overhead, the proposed method may facilitate broader adoption of principled second-order optimization techniques in practice. As with reinforcement learning methods in general, potential downstream applications span a wide range of domains and should be deployed responsibly, particularly in efficiency-critical settings. This paper focuses on algorithmic contributions and does not involve human subjects or sensitive data.

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

# A. Gradient Calculation in the Likelihood Example

For the likelihood example, the PG and NPG are given in the closed forms as follows:

$$\underbrace{\nabla_\theta \mathbb{E}_x \left[\log \mathcal{N}(x|\theta)\right] = \mathbb{E}_x \begin{bmatrix} \frac{(x-\theta_1)}{\exp(2\theta_2)} \\ -1 + \frac{(x-\theta_1)^2}{\exp(2\theta_2)} \end{bmatrix}}_{\text{Vanilla gradient}}, \quad \underbrace{F^{-1}(\theta)\nabla_\theta \mathbb{E}_x \left[\log \mathcal{N}(x|\theta)\right] = \mathbb{E}_x \begin{bmatrix} (x-\theta_1) \\ -\frac{1}{2} + \frac{(x-\theta_1)^2}{2\exp(2\theta_2)} \end{bmatrix}}_{\text{Natural gradient}}$$

Specifically, with parameterization $\theta_1 = \mu$ and $\theta_2 = \log \sigma$, we have

$$\mathcal{N}(x|\mu,\sigma;\theta) = \frac{1}{\sqrt{2\pi}\sigma} \exp\left[-\frac{(x-\mu)^2}{2\sigma^2}\right] = \frac{1}{\sqrt{2\pi}\exp(\theta_2)} \exp\left[-\frac{(x-\theta_1)^2}{2\exp(2\theta_2)}\right], \tag{18}$$

$$\log \mathcal{N}(x|\mu,\sigma;\theta) = -\log\sqrt{2\pi} - \theta_2 - \frac{(x-\theta_1)^2}{2\exp(2\theta_2)}. \tag{19}$$

The gradient of log probability is given as

$$\nabla_\theta \log \mathcal{N}(x|\theta) = \begin{bmatrix} \exp(-2\theta_2)(x-\theta_1) \\ -1 + \exp(-2\theta_2)(x-\theta_1)^2 \end{bmatrix} = \begin{bmatrix} \frac{(x-\theta_1)}{\sigma^2} \\ -1 + \frac{(x-\theta_1)^2}{\sigma^2}. \end{bmatrix} \tag{20}$$

Accordingly, the Fisher matrix at point $x$ is given

$$F(x,\theta) = \begin{bmatrix} \frac{(x-\theta_1)^2}{\sigma^4} & \frac{(x-\theta_1)^3}{\sigma^4} - \frac{(x-\theta_1)}{\sigma^2} \\ \frac{(x-\theta_1)^3}{\sigma^4} - \frac{(x-\theta_1)}{\sigma^2} & 1 - \frac{2(x-\theta_1)^2}{\sigma^2} + \frac{(x-\theta_1)^4}{\sigma^4}. \end{bmatrix} \tag{21}$$

Thus, the full Fisher matrix is

$$F(\theta) = \mathbb{E}_{x\sim\mathcal{N}(x|\mu,\sigma,\theta)} \begin{bmatrix} \frac{(x-\theta_1)^2}{\sigma^4} & \frac{(x-\theta_1)^3}{\sigma^4} - \frac{(x-\theta_1)}{\sigma^2} \\ \frac{(x-\theta_1)^3}{\sigma^4} - \frac{(x-\theta_1)}{\sigma^2} & 1 - \frac{2(x-\theta_1)^2}{\sigma^2} + \frac{(x-\theta_1)^4}{\sigma^4} \end{bmatrix} \tag{22}$$

$$= \begin{bmatrix} \mathbb{E}_{x\sim\mathcal{N}(x|\mu,\sigma;\theta)}\left[\frac{(x-\theta_1)^2}{\sigma^4}\right] & \mathbb{E}_{x\sim\mathcal{N}(x|\mu,\sigma;\theta)}\left[\frac{(x-\theta_1)^3}{\sigma^4} - \frac{(x-\theta_1)}{\sigma^2}\right] \\ \mathbb{E}_{x\sim\mathcal{N}(x|\mu,\sigma;\theta)}\left[\frac{(x-\theta_1)^3}{\sigma^4} - \frac{(x-\theta_1)}{\sigma^2}\right] & \mathbb{E}_{x\sim\mathcal{N}(x|\mu,\sigma;\theta)}\left[1 - \frac{2(x-\theta_1)^2}{\sigma^2} + \frac{(x-\theta_1)^4}{\sigma^4}\right] \end{bmatrix} \tag{23}$$

$$= \begin{bmatrix} \frac{1}{\sigma^2} & 0 \\ 0 & 2 \end{bmatrix} = \begin{bmatrix} \exp(-2\theta_2) & 0 \\ 0 & 2 \end{bmatrix}. \tag{24}$$

The transition from Equation (23) to Equation (24) is based on the fact that if $x$ has a normal distribution $\mathcal{N}(x|\mu,\sigma)$, the non-central moments exist for any non-negative integer $p$ and are given as follows:

$$\mathbb{E}_x \left[(x-\mu)^p\right] = \begin{cases} 0 & \text{if } p \text{ is odd,} \\ \sigma^p (p-1)!! & \text{if } p \text{ is even.} \end{cases} \tag{25}$$

For a diagonal Fisher matrix, its inverse exists since $\exp(2\theta_2) > 0$ and is given below

$$F^{-1}(\theta) = \begin{bmatrix} \sigma^2 & 0 \\ 0 & \frac{1}{2} \end{bmatrix} = \begin{bmatrix} \exp(2\theta_2) & 0 \\ 0 & \frac{1}{2} \end{bmatrix}. \tag{26}$$

Thus, the vanilla gradient direction for the parameter update is

$$\nabla_\theta \mathbb{E}_x \left[\log p(x|\theta)\right] = \mathbb{E}_x \begin{bmatrix} \frac{(x-\theta_1)}{\exp(2\theta_2)} \\ -1 + \frac{(x-\theta_1)^2}{\exp(2\theta_2)} \end{bmatrix}, \tag{27}$$

and the natural gradient direction is

$$F^{-1}(\theta)\mathbb{E}_x \left[\nabla_\theta p(x|\theta)\right] = \mathbb{E}_x \left[F^{-1}(\theta)\nabla_\theta p(x|\theta)\right] = \mathbb{E}_x \begin{bmatrix} (x-\theta_1) \\ -\frac{1}{2} + \frac{(x-\theta_1)^2}{2\exp(2\theta_2)} \end{bmatrix} \tag{28}$$

For damped Fisher $\lambda I + F(\theta)$, its inverse is given by

$$(\lambda I + F(\theta))^{-1} = \begin{bmatrix} \frac{\sigma^2}{1+\lambda\sigma^2} & 0 \\ 0 & \frac{1}{2+\lambda} \end{bmatrix} = \begin{bmatrix} \frac{\exp(2\theta_2)}{1+\lambda\exp(2\theta_2)} & 0 \\ 0 & \frac{1}{2+\lambda} \end{bmatrix}. \tag{29}$$

The regularized natural gradient direction is

$$(\lambda I + F(\theta))^{-1} \, \mathbb{E}_x\left[\nabla_\theta p(x|\theta)\right] = \mathbb{E}_x\left[ \begin{matrix} \frac{x-\theta_1}{1+\lambda\exp(2\theta_2)} \\ -\frac{1}{2+\lambda} + \frac{(x-\theta_1)^2}{(2+\lambda)\exp(2\theta_2)} \end{matrix} \right] \tag{30}$$

To compute the gradients, we randomly sample 2,000 data points from $\mathcal{N}(0,1)$.

# B. Convergence Results of Randomized Advantage Transformation

Consider the regularized least-squares objective

$$\min_{g\in\mathbb{R}^p} \quad \frac{1}{2}\|y - Hg\|_2^2 + \frac{\lambda}{2}\|g\|_2^2, \tag{31}$$

with unique solution

$$g^* := (H^\top H + \lambda I)^{-1} H^\top y. \tag{32}$$

At iteration $j$, RAT samples a minibatch $\tau_j$ and performs the update

$$g_{j+1} = g_j + H_{\tau_j}^\top (\lambda I + H_{\tau_j} H_{\tau_j}^\top)^{-1}\left(y_{\tau_j} - H_{\tau_j} g_j\right). \tag{33}$$

Define

$$P_\tau := H_\tau^\top(\lambda I + H_\tau H_\tau^\top)^{-1} H_\tau, \qquad S_\tau := H_\tau^\top(\lambda I + H_\tau H_\tau^\top)^{-1}. \tag{34}$$

Let the estimation error be

$$e_j := g_j - g^*. \tag{35}$$

**Lemma 2.** *For any minibatch $\tau$, the matrix $P_\tau$ satisfies*

$$0 \preceq P_\tau \preceq I, \qquad P_\tau^2 \preceq P_\tau.$$

*Proof.* Let $H_\tau = U\Sigma V^\top$ be the singular value decomposition. Then

$$P_\tau = V\Sigma^\top(\lambda I + \Sigma\Sigma^\top)^{-1}\Sigma V^\top = V \operatorname{diag}\left(\frac{\sigma_i^2}{\lambda + \sigma_i^2}\right) V^\top.$$

Each eigenvalue lies in $[0,1)$, implying $0 \preceq P_\tau \preceq I$. Since $x^2 \le x$ for $x \in [0,1]$, we also have $P_\tau^2 \preceq P_\tau$. $\qquad\square$

**Lemma 3.** *Under the assumptions of full column rank of $H$ and full data coverage,*

$$\mu := \lambda_{\min}(\mathbb{E}[P_\tau]) > 0.$$

*Proof.* Recall

$$P_\tau = H_\tau^\top(\lambda I + H_\tau H_\tau^\top)^{-1} H_\tau \succeq 0, \quad \text{and} \quad v^\top P_\tau v = 0 \iff H_\tau v = 0.$$

We prove $\mathbb{E}[P_\tau] \succ 0$ by showing that no nonzero vector $v$ can satisfy $H_\tau v = 0$ almost surely. Fix any $v \ne 0$. Since $H$ has full column rank, we have $Hv \ne 0$, hence there exists at least one index $i$ such that $h_i^\top v \ne 0$. By full data coverage, $\mathbb{P}(i \in \tau) > 0$. On the event $\{i \in \tau\}$, the submatrix $H_\tau$ contains row $h_i^\top$, and thus $H_\tau v \ne 0$ (because its $i$-th component equals $h_i^\top v \ne 0$). Therefore,

$$\mathbb{P}(H_\tau v \ne 0) \ge \mathbb{P}(i \in \tau) > 0.$$

Consequently, $H_\tau v = 0$ cannot hold almost surely for any nonzero $v$.

Since $(\lambda I + H_\tau H_\tau^\top)^{-1} \succ 0$, we have $v^\top P_\tau v > 0$ whenever $H_\tau v \ne 0$, and hence

$$v^\top \mathbb{E}[P_\tau] v = \mathbb{E}[v^\top P_\tau v] > 0 \quad \text{for all } v \ne 0.$$

Thus $\mathbb{E}[P_\tau] \succ 0$, which implies $\mu = \lambda_{\min}(\mathbb{E}[P_\tau]) > 0$. $\qquad\square$

We first analyze the idealized case where minibatch targets are exact. For all minibatches $\tau$, $\boldsymbol{y}_\tau = \boldsymbol{H}_\tau \boldsymbol{g}^*$.

**Theorem 3** (Linear convergence of RAT). *Assume minibatches $\tau_j$ are sampled i.i.d. from an arbitrary distribution. Define $\mu := \lambda_{\min}(\mathbb{E}[\boldsymbol{P}_\tau])$, then*

$$\mathbb{E}\|\boldsymbol{g}_j - \boldsymbol{g}^*\|_2^2 \leq (1-\mu)^j \|\boldsymbol{g}_0 - \boldsymbol{g}^*\|_2^2. \tag{36}$$

*Proof.* Using the noise-free assumption, the update Equation (33) becomes

$$\boldsymbol{g}_{j+1} = \boldsymbol{g}_j - \boldsymbol{P}_{\tau_j}(\boldsymbol{g}_j - \boldsymbol{g}^*).$$

Subtracting $\boldsymbol{g}^*$ yields

$$\boldsymbol{e}_{j+1} = (\boldsymbol{I} - \boldsymbol{P}_{\tau_j})\boldsymbol{e}_j.$$

Conditioned on $\boldsymbol{e}_j$ and $\tau_j$,

$$\|\boldsymbol{e}_{j+1}\|_2^2 = \boldsymbol{e}_j^\top (\boldsymbol{I} - 2\boldsymbol{P}_{\tau_j} + \boldsymbol{P}_{\tau_j}^2)\boldsymbol{e}_j.$$

Since $\boldsymbol{P}_{\tau_j}^2 \preceq \boldsymbol{P}_{\tau_j}$,

$$\boldsymbol{I} - 2\boldsymbol{P}_{\tau_j} + \boldsymbol{P}_{\tau_j}^2 \preceq \boldsymbol{I} - \boldsymbol{P}_{\tau_j},$$

and therefore

$$\|\boldsymbol{e}_{j+1}\|_2^2 \leq \|\boldsymbol{e}_j\|_2^2 - \boldsymbol{e}_j^\top \boldsymbol{P}_{\tau_j} \boldsymbol{e}_j.$$

Taking conditional expectation,

$$\mathbb{E}[\|\boldsymbol{e}_{j+1}\|_2^2 \mid \boldsymbol{e}_j] \leq \|\boldsymbol{e}_j\|_2^2 - \boldsymbol{e}_j^\top \mathbb{E}[\boldsymbol{P}_\tau]\boldsymbol{e}_j.$$

Since $\boldsymbol{e}_j^\top \mathbb{E}[\boldsymbol{P}_\tau]\boldsymbol{e}_j \geq \mu\|\boldsymbol{e}_j\|_2^2$,

$$\mathbb{E}[\|\boldsymbol{e}_{j+1}\|_2^2 \mid \boldsymbol{e}_j] \leq (1-\mu)\|\boldsymbol{e}_j\|_2^2.$$

Iterating proves the claim. □

We now consider stochastic targets, as in reinforcement learning. For each minibatch $\tau$, $\boldsymbol{y}_\tau = \boldsymbol{H}_\tau \boldsymbol{g}^* + \boldsymbol{\xi}_\tau$, where the noise $\boldsymbol{\xi}_\tau$ satisfies $\mathbb{E}[\boldsymbol{\xi}_\tau \mid \tau] = \boldsymbol{0}$.

**Theorem 4** (Convergence with error floor). *Define $\eta^2 := \mathbb{E}[\|\boldsymbol{S}_\tau \boldsymbol{\xi}_\tau\|_2^2]$, then*

$$\mathbb{E}\|\boldsymbol{g}_j - \boldsymbol{g}^*\|_2^2 \leq (1-\mu)^j \|\boldsymbol{g}_0 - \boldsymbol{g}^*\|_2^2 + \frac{\eta^2}{\mu}. \tag{37}$$

*Proof.* Substituting the stochastic model into Equation (33) yields

$$\boldsymbol{e}_{j+1} = (\boldsymbol{I} - \boldsymbol{P}_{\tau_j})\boldsymbol{e}_j + \boldsymbol{S}_{\tau_j}\boldsymbol{\xi}_{\tau_j}.$$

Squaring and taking conditional expectation, the cross term vanishes since $\mathbb{E}[\boldsymbol{\xi}_{\tau_j} \mid \tau_j] = \boldsymbol{0}$, giving

$$\mathbb{E}[\|\boldsymbol{e}_{j+1}\|_2^2 \mid \boldsymbol{e}_j] \leq (1-\mu)\|\boldsymbol{e}_j\|_2^2 + \eta^2.$$

Unrolling the resulting recursion completes the proof. □

## C. Implementation Details

We implemented RAT using the per-sample gradients feature in PyTorch[3]. Specifically, we first compute the per-sample gradients of the policy network's outputs with respect to its parameters using the built-in function `torch.func.grad`, `torch.func.vmap` and `torch.func.functional_call`, and then flat these per-sample gradients to compute $\boldsymbol{H}$. When computing $\boldsymbol{H}\boldsymbol{H}^\top$, we average over the samples in the mini-batch. We use `torch.linalg.solve` to solve the linear system involving the damped Fisher matrix $(\lambda I + \boldsymbol{H}\boldsymbol{H}^\top)$, and thus apply the advantage transformation. This is a faster and more numerically stable way than performing the computations separately.

Besides, we also incorporated the following training techniques:

---

[3] https://docs.pytorch.org/tutorials/intermediate/per_sample_grads.html

**Observation normalization.** We normalize the observations with running mean and standard deviation as in (Schulman et al., 2017b) for all the MuJoCo tasks and set the clip range to $[-5, 5]$. For tasks with image observations, we normalize the pixel values to $[0, 1]$ by dividing them by 255 and then normalize each pixel value with 0.5 mean and 0.5 standard deviation (to ensure the pixel values are in the range of $[-1, 1]$). We also stack the last three frames as the input to the policy network. We found that observation normalization is crucial for stabilizing training, especially for tasks in the Mujoco suite. Note that after observation normalization, we re-evaluate the action distribution's mean and variance, i.e., $\mu(s)$ and $\sigma(s)$, to ensure that the action distribution is consistent with the normalized observations.

**Advantage normalization.** We use the Generalized Advantage Estimation (GAE) (Schulman et al., 2016) to compute the advantage estimates. We then normalize the advantage estimates to have zero mean and unit standard deviation within each batch as in (Schulman et al., 2017b). We found that advantage normalization is important for stabilizing training, especially when using high learning rates.

**PopArt value normalization.** We use the PopArt normalization technique (Hessel et al., 2019) to normalize the value function targets. Specifically, we maintain running estimates of the mean $\mu$ and standard deviation $\sigma$ of the value function targets and normalize the targets as $(G_t - \mu)/\sigma$. We also adjust the parameters of the value network to account for the change in normalization following the procedure described in (Hessel et al., 2019). We set the decay rate for the running estimates to 0.99999. One slight improvement we made in our implementation is that we correct the bias in the running estimates of the mean and standard deviation by dividing the estimates by $(1 - \text{decay}^t)$ at time step $t$, similar to Adam (Kingma & Ba, 2015).

**Gradient clipping.** We clip the gradient norm to be at most 0.5 when updating the shared policy and value networks. If the gradient norm exceeds this threshold, we scale down the gradient to have a norm of 0.5. We also tried the Fisher norm clipping technique proposed in (Ba et al., 2017), but found that $l2$ norm clipping works equally well in our experiments. When the actor and critic networks are separate, we apply gradient clipping to the policy network with a threshold of 0.5, and to the value network with a threshold of 5.0.

**Action squashing.** For environments with bounded action spaces, we apply a squashing function (tanh) to the actions sampled from the Gaussian policy to ensure that the actions lie within the valid range. Different from the procedure described in (Haarnoja et al., 2018), we do not adjust the log-probability of the actions to account for the squashing transformation. This is because the policy ratios, KL divergences, and Fisher matrix are all invariant to such transformations, as long as the transformation is differentiable and invertible.

**Ratio clamping.** When computing the policy ratios, we clamp ratios to be within $[10^{-1}, 10^1]$ to avoid numerical instability.

## C.1. RAT in Shared Actor-Critic

In Actor-Critic methods, the actor and critic often share a common neural architecture (Mnih et al., 2016; Wu et al., 2017). When parameters are shared, we follow Wu et al. (2017) and estimate the joint natural policy gradients for the actor and critic. Specifically, we model the value output as a Gaussian distribution with fixed variance $\sigma^2$, i.e., $p(v|s) = \mathcal{N}(v; V(s), \sigma^2)$, where $v$ denotes a target value obtained from Monte-Carlo rollouts or Temporal Difference (TD) methods (Sutton & Barto, 2018). The critic is trained by maximizing the log-likelihood of this distribution, and the Fisher matrix for the critic is defined with respect to the corresponding log-likelihood. In practice we set $\sigma$ to 1 without loss of generality, yielding $\log p(v|s) \propto -\|v - V(s)\|^2$.

$$\max \mathbb{E}_{v \sim q}\left[\log p(v|s)\right] = \mathbb{E}_{v \sim q}\left[-\|v - V(s)\|^2\right]$$

Under this formulation, the score function for the joint distribution $p(a, v|s)$ factorizes as

$$\nabla_{\boldsymbol{\theta}} \log p(a, v|s, \boldsymbol{\theta}) = \nabla_{\boldsymbol{\theta}} \log \pi(a|s; \boldsymbol{\theta}) + \nabla_{\boldsymbol{\theta}} \log p(v|s; \boldsymbol{\theta}),$$

which is used to construct the matrix $\boldsymbol{H}$ in RAT. Following Wu et al. (2017), we sample the network outputs independently for the actor and critic, and inject unit-variance Gaussian noise to the value outputs.

To apply RAT to the critic, we introduce a pseudo advantage for the value loss, such as an all-ones vector $\mathbf{1}$ with the same size as the mini-batch. Applying RAT to this pseudo advantage yields $\tilde{w}(s)$ for each state, which can be interpreted as the

natural gradient update direction for the critic. The resulting joint loss for optimizing shared actor-critic networks is

$$\mathcal{L}(\boldsymbol{\theta}) = -\mathbb{E}\left[\frac{\pi(a|s;\boldsymbol{\theta})}{\pi_{\text{old}}(a|s)}\tilde{A}(s,a)\right] + \mathbb{E}\left[\tilde{w}(s)\|v - V(s;\boldsymbol{\theta})\|^2\right]$$

where $\tilde{A}(s,a)$ is the transformed advantage for the actor, and $\tilde{w}(s)$ is the transformed pseudo advantage for the critic. At each iteration, RAT is applied jointly to transform both the actor advantage for the critic pseudo advantage, enabling a unified and stable natural-gradient update for shared acrtor-critic networks.

---

**Algorithm 1** Randomized Advantage Transformation (RAT)

---

1: **Input:** Initial policy parameters $\boldsymbol{\theta}_0$, batch size $B$, total iterations $K$
2: **for** $k = 0$ to $K - 1$ **do**
3:     Collect on-policy samples: $\pi_{\text{old}}(a|s) \leftarrow \pi(a|s;\boldsymbol{\theta}_k)$
4:     Randomly partition $\mathcal{D}_k$ into batches of size $B$
5:     **for** each batch $\tau_j$ in $\mathcal{D}_k$ **do**
6:         Form $\boldsymbol{H}_{\tau_j}$ using samples in $\tau_j$
7:         Apply RAT:

$$\tilde{A}_j(s,a) = \left[(\lambda\boldsymbol{I} + \boldsymbol{H}_{\tau_j}\boldsymbol{H}_{\tau_j}^\top)^{-1}\left(\boldsymbol{y}_{\tau_j} - \boldsymbol{H}_{\tau_j}\boldsymbol{g}_{j-1}\right)\right]_{(s,a)}$$

8:         Estimate gradient via a single backpropagation of the following loss:

$$\boldsymbol{g}_j = -\frac{\partial}{\partial\boldsymbol{\theta}}\mathbb{E}\left[\frac{\pi(a|s;\boldsymbol{\theta})}{\pi_{\text{old}}(a|s)}\tilde{A}_j(s,a)\right]$$

9:         Apply gradient clipping $\alpha_k := \min\left(\eta, \frac{\nu}{\|\boldsymbol{g}_j\|_2}\right)$.
10:        Update policy parameters: $\boldsymbol{\theta} \leftarrow \boldsymbol{\theta} + \alpha_k\boldsymbol{g}_j$
11:     **end for**
12: **end for**
13: **Output:** Policy parameters $\boldsymbol{\theta}$

---

### C.2. Gap between Theoretical Analysis and Practical Implementation

The theorems analyze RAT as a fixed-policy linear system solver, while Algorithm 1 interleaves these updates with policy optimization, resulting in a time-varying sequence of systems. This makes the linear system non-stationary across inner iterations. The resulting algorithm is closer to "multiple PPO-like updates per rollout with curvature-corrected advantages" than to an iterative linear solver converging to a fixed target. To clarify the gap formally, let $\boldsymbol{g}_t^*$ denote the solution of the regularized least-squares problem defined by the current policy $\boldsymbol{\theta}_t$, and define the tracking error

$$\boldsymbol{e}_t := \|\boldsymbol{g}_t - \boldsymbol{g}_t^*\|. \tag{38}$$

At iteration $t$, RAT performs an update yielding $\boldsymbol{g}_{t+1}$. The fixed-system analysis ( Theorem 1) implies a contraction:

$$\|\boldsymbol{g}_{t+1} - \boldsymbol{g}_t^*\| \leq \rho\|\boldsymbol{g}_t - \boldsymbol{g}_t^*\| = \rho\boldsymbol{e}_t, \tag{39}$$

where $\rho = 1 - \mu < 1$. After the policy update $\boldsymbol{\theta}_t \mapsto \boldsymbol{\theta}_{t+1}$, the target solution shifts. Under standard smoothness assumptions on $\boldsymbol{H}$ and $\boldsymbol{y}$, the solution map $\boldsymbol{\theta} \mapsto \boldsymbol{g}^*(\boldsymbol{\theta})$ is Lipschitz:

$$\|\boldsymbol{g}^*(\boldsymbol{\theta}_{t+1}) - \boldsymbol{g}^*(\boldsymbol{\theta}_t)\| \leq L\|\boldsymbol{\theta}_{t+1} - \boldsymbol{\theta}_t\|. \tag{40}$$

Combining these yields:

$$\boldsymbol{e}_{t+1} \leq \rho\boldsymbol{e}_t + L\|\boldsymbol{\theta}_{t+1} - \boldsymbol{\theta}_t\| \tag{41}$$

Unrolling:

$$\boldsymbol{e}_t \leq \rho^t\boldsymbol{e}_0 + L\sum_{s=0}^{t-1}\rho^{t-1-s}\|\boldsymbol{\theta}_{s+1} - \boldsymbol{\theta}_s\| \tag{42}$$

This shows that RAT can be interpreted as a *contractive solver tracking a slowly varying sequence of systems*. Under our settings (small learning rates and gradient clipping), the drift term remains small, yielding a bounded steady-state error of order

$$\mathcal{O}(\max_t \|\boldsymbol{\theta}_{t+1} - \boldsymbol{\theta}_t\|) \tag{43}$$

This is a standard tracking bound for contractive iterative methods applied to slowly varying systems and provides a principled justification for the interleaved algorithm.

Specifically, while the Lipschitz constant $L$ is not directly measurable, the step size $\|\boldsymbol{\theta}_{t+1} - \boldsymbol{\theta}_t\|$ is explicitly controlled by the learning rate and gradient clipping. In particular, we use $\ell_2$ gradient norm clipping at $0.5$ together tiwth a learning rate $0.05$ for the policy network, which ensures that each parameter update is bounded by at most $0.5 \times 0.05 = 0.025$ in $\ell_2$ norm. This keeps the drift term small throughout training up to the unknown constant $L$. More broadly, this is precisely why gradient clipping and small learning rates are important in our implementation: they ensure that the target system evolves slowly enough for the tracking interpretation to be meaningful.

## D. Hyperparameters

We summarize the hyperparameters used in our experiments in Table 3.

*Table 3.* Common hyperparameters used in our experiments.

| Hyperparameter | Value |
| --- | --- |
| Discount factor $\gamma$ | 0.99 |
| GAE parameter $\lambda$ | 0.95 |
| Damping factor $\lambda$ | $10^{-1}$ |
| Mini-batch size | 1024 |
| PPO clipping parameter $\epsilon$ | 0.2 |
| Number of epochs per update | 8 |
| Number of steps per update | 256 * 32 |
| Gradient clipping threshold (policy) | 0.5 |
| Gradient clipping threshold (value) | 5.0 |
| PopArt decay rate | 0.99999 |
| Observation normalization clip range | $[-5, 5]$ |
| Entropy coefficient | 0 |

*Table 4.* Hyperparameters for RAT.

| Hyperparameter | Value |
| --- | --- |
| pi lr | 0.05 |
| vf lr | 0.001 |
| lr for shared network | 0.1 |
| damping for MLP | 0.1 |
| damping for CNN & ResNet | 0.5 |

*Table 5.* Hyperparameters for KFAC (adopted from (George et al., 2018)).

| Hyperparameter | Value |
| --- | --- |
| lr | 0.001 |
| momentum | 0.9 |
| stat decay | 0.95 |
| damping | 0.001 |
| kl clip | 0.001 |
| weight decay | 0 |
| TCov | 1 |
| TInv | 10 |
| batch averaged | True |

*Table 6.* Hyperparameters for PPO.

| Hyperparameter | Value |
| --- | --- |
| pi lr | 0.01 |
| vf lr | 0.001 |
| lr for shared networks | 0.001 |
| momentum | 0.9 |
| stat decay | 0.95 |
| damping | 0.001 |
| kl clip | 0.001 |
| weight decay | 0 |
| TCov | 1 |
| TInv | 10 |
| batch averaged | True |

Architecture details:

- For MuJoCo tasks, we use a two-layer MLP with 256 hidden units per layer and tanh activations for both the policy and value networks.

- For Procgen tasks, we use the same ResNet architecture as in (Cobbe et al., 2020): four residual blocks with 16, 32, and 32 filters respectively, followed by a fully connected layer with 256 units. We use ReLU activations after each layer.

# E. Additional Experimental Results

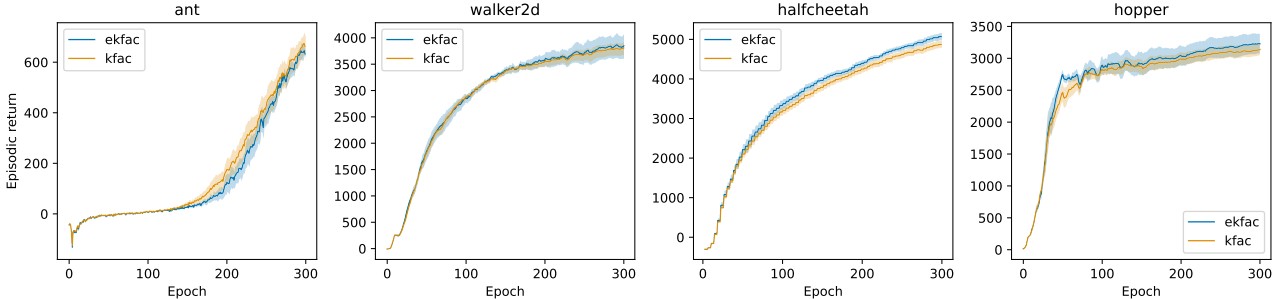

*Figure 6.* Comparing KFAC and EKFAC on Continuous Control Tasks. EKFAC performs similar to KFAC in most tasks.

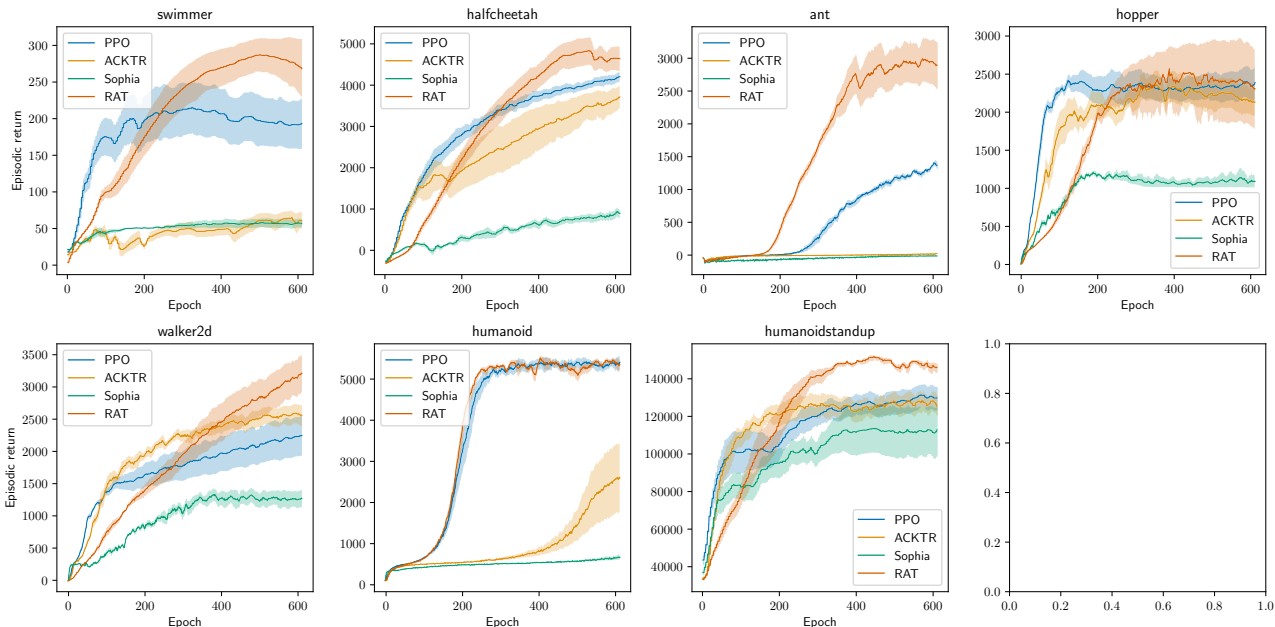

*Figure 7.* Optimizing MLP Policies on Continuous Control Tasks with Shared Actor-Critic Networks. RAT outperforms KFAC and FVP+CG in most tasks. The shaded region denotes the standard deviation over 5 random seeds.

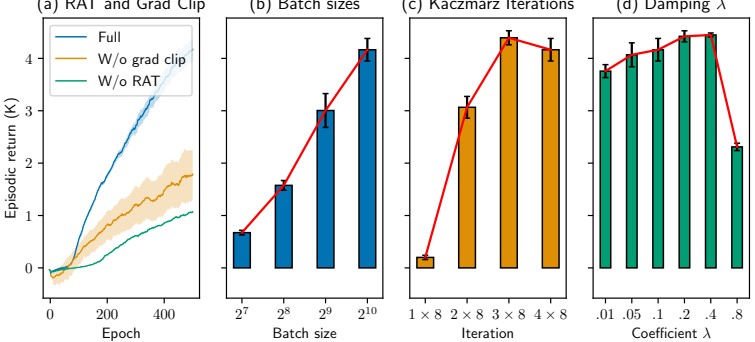

*Figure 8.* Ablation study and sensitivity analysis of RAT on Ant.

