# OpenReview forum: "Randomized Advantage Transformation (RAT): Computing Natural Policy Gradients via Direct Backpropagation"
_ICML.cc/2026/Conference — ICML 2026 regular_

### Official Review · Reviewer_yPHb · 2026-02-18

**Soundness:** 4
**Presentation:** 4
**Significance:** 4
**Originality:** 4
**Overall Recommendation:** 6
**Confidence:** 4

**Summary:**

This paper proposes Randomized Advantage Transformation: an efficient method for computing natural policy gradients by transforming the advantage, which they prove is equivalent to using the inverse Fisher matrix without ever having to calculate it. Advantage transformation does not rely on the number of parameters, like the Fisher matrix, making the method architecture-agnostic. Advantage is computed using randomized linear solvers. Natural policy gradients are computed using standard backpropagation on a PPO-like loss, which is the ratio of the probability of the current policy selecting an action vs the old policy, times the computed advantage. Proofs for this method are provided, as are theoretical guarantees. Empirical results are performed on MuJoCo and Procgen environments.

**Compliance With Llm Reviewing Policy:**

Affirmed.

**Final Justification:**

The rebuttal addressed my concerns and I have raised my score as indicated in response.

**Key Questions For Authors:**

See weaknesses 1 and 2. Both weaknesses touch on limitations in the empirical studies. Improving these, or justifying the choices made, would increase the significance of the work.

**Limitations:**

Yes

**Strengths And Weaknesses:**

Strengths:
1. The paper has a strong theoretical backing and provides solid proofs.
2. Empirical results back up theoretical claims. While RAT does not always outperform other NPG methods and is more expensive than some methods, it achieves consistently high performance while balancing costs, and appears to reach high performance in relatively few epochs. The theoretical guarantees are backed up by RAT’s consistency, even when RAT is outperformed by competing methods.
3. The algorithm and hyperparameters are provided in the appendix for reproducibility.
4. The theoretical work is very dense by necessity, but the authors do a good job guiding the reader through.

Weaknesses:
1. The experiments in Figure 3 show each method with a fixed computation budget of 1250 epochs, but many of the methods appear not to have converged at that point. For example, Sophia in the humanoid graph clearly has an upward trajectory by the 1250th epoch. As does KFAC in halfcheetah, which is already outperforming RAT. Is there a specific justification for this computation budget? If not, it does not seem a fair comparison to cut off training before competing methods have converged.
2. The experiments were performed with a relatively small actor network, with only two linear layers. Since RAT does not rely on the number of parameters for advantage computation, it should be suitable for more complex networks. Demonstrating RAT’s empirical performance against these other methods with a larger network (even just a few layers deeper) would strengthen the empirical results and show RAT’s architecture agnosticity. If the other methods struggle to scale with higher parameter counts, this combined with the current experiments would only expand the use cases and strengthen the significance of the method.

---

> ### Author Rebuttal · Authors · 2026-03-28
>
> We thank the reviewer for the detailed and constructive feedback. We address each concern below and will incorporate the suggested clarifications in the revision.
>
> ---
>
> ### 1. Fixed training budget (1250 epochs)
> We appreciate this point. Our goal in using a fixed training budget was to ensure a **controlled and consistent evaluation protocol** across all methods, rather than tuning the horizon individually.
>
> In our setup, 1250 epochs correspond to approximately **10 million environment steps, which is a standard budget in continuous control benchmarks**. We chose this budget to reflect a practical training regime, where computational constraints are fixed across methods.
>
> While some baselines (e.g., KFAC, Sophia) continue improving beyond this point, our main conclusions are based on:
> + performance within a fixed compute budget, and
> + sample efficiency and stability during training
>
> In this regime, RAT consistently achieves strong and stable performance, often reaching high returns earlier than competing methods.
>
> We agree that extended training could provide additional insights into asymptotic behavior. We will clarify this limitation in the paper and avoid overinterpreting results in the non-converged regime.
>
> ---
>
> ### 2. Network size / architecture scaling
> We agree that demonstrating scalability to larger networks is important. We would like to clarify that the paper already includes experiments beyond 2-layer MLPs.
>
> In particular, Sec. 5.3 evaluates RAT on **ResNet-based policies** (following Espeholt et al., 2018), consisting of **3 convolutional residual blocks with 5 convolutional layers each (15 convolutional layers in total)** on Procgen environments. These experiments demonstrate that RAT scales effectively to **deep, high-dimensional architectures**, well beyond standard MLP settings.
>
> Importantly, in these experiments RAT maintains **strong and stable performance**, while we observe that **KFAC-based methods (e.g., ACKTR) tend to struggle with increasing parameter counts**, often exhibiting instability or degraded performance. This further supports our claim that **RAT is architecture-agnostic in practice**, as its core computation depends on minibatch structure rather than model size.
>
> To make this clearer, we will revise the paper to explicitly highlight these ResNet results as evidence of scalability, and emphasize the contrast with existing natural-gradient methods.
>
> ---
>
> We appreciate your positive feedback on Soundness, Presentation and Originality.

---

> > ### Author Rebuttal · Reviewer_yPHb · 2026-03-31
> >
> > This rebuttal answered my concerns with the empirical studies. My scaling concern was already answered in the paper, as stated in rebuttal point 2, and the response to point 1 is fair. The only change to the paper I request in response is for a clear statement for the reasoning behind the fixed training budget. I've increased by significance, overall score, and confidence in response.

---

> > > ### Author Response · Authors · 2026-03-31
> > >
> > > Thank you! We really appreciate your positive feedback and support!

---

### Official Review · Reviewer_x9nY · 2026-03-12

**Soundness:** 4
**Presentation:** 3
**Significance:** 3
**Originality:** 3
**Overall Recommendation:** 5
**Confidence:** 4

**Summary:**

This paper presents Randomized Advantage Transformation (RAT), a novel method for estimating Tikhonov-regularized natural policy gradients (NPG) that bypasses the traditional computational bottlenecks of explicit Fisher matrix construction or conjugate-gradient solvers. By leveraging the Woodbury formula, the authors reformulate the regularized NPG update into a vanilla policy gradient form with a transformed advantage, effectively absorbing curvature information into a sample-dependent modification. This transformation is efficiently approximated via randomized block Kaczmarz iterations on on-policy mini-batches, allowing the gradient to be computed through standard backpropagation and making the approach entirely architecture-agnostic. Theoretically, the authors establish that RAT converges linearly to the regularized NPG under standard assumptions. Empirically, RAT demonstrates robust performance, matching or exceeding established second-order methods like KFAC and FVP+CG across continuous control (MuJoCo) and high-dimensional visual (Procgen) benchmarks, while offering a more favorable trade-off between wallclock efficiency and optimization stability.

**Compliance With Llm Reviewing Policy:**

Affirmed.

**Final Justification:**

The rebuttal has clarified my concern about the empirical performance under certain conditions. I have raised my score.

**Key Questions For Authors:**

1. Validity of Assumptions in Continuous Spaces
Does the empirical rank of $H$ typically hold for Assumptions 1 and 2 during training, or do you observe "rank collapse" in certain regions of the state space?. Furthermore, how does the algorithm's performance degrade if the state-action coverage in a single rollout is poor?

2. Empirical effect of bounded action space
While the direction of the natural gradient is invariant, the saturation of the tanh function (where gradients become very small) can lead to numerical instability in the advantage transformation. If the policy is pushed into the "flat" regions of the tanh, will the empirical Fisher matrix estimated by RAT become ill-conditioned? Is this the reason why gradient clipping is important?

Overall it is a solid paper with theoritical grounding and good empirical performance.

**Limitations:**

Yes.

**Strengths And Weaknesses:**

Strengths:
1. Algorithmic Simplicity and Generality:
2. Good empirical performance.
3. Strong theoretical results.

Weakness:
1. The methods are not scalable to larger batch sizes and are sensitive to the sampling quality.
2. More baselines should be considered.

---

> ### Author Rebuttal · Authors · 2026-03-28
>
> We thank the reviewer for the detailed and constructive feedback. We address each concern below and will incorporate the suggested clarifications in the revision.
>
> ---
>
> ### 1. Rank assumption
> Assumption 1 applies to the **full matrix** $H$, not to individual minibatches. In practice, minibatch matrices $H_\tau$ can be low-rank. The Tikhonov damping term $\lambda > 0$ ensures that $(\lambda I + H_\tau H_\tau^\top)$ is always invertible, even when $H_\tau$ is rank-deficient.
> Poor state-action coverage affects the convergence rate through the constant $\mu=\lambda_\min(E[P_\tau])$, but does not invalidate the method. We will clarify this distinction in the revision.
>
> ---
>
> ### 2. Bounded action space / tanh saturation
> You are correct that saturated tanh nonlinearities can reduce gradient magnitudes and potentially worsen conditioning. In our implementation, however, the tanh squashing is **not part of the computational graph used for gradient estimation**.
>
> Specifically, the policy outputs **unsquashed Gaussian actions**, and tanh is applied only when interacting with the environment. For on-policy updates, we store and use the **pre-squashed actions** when computing log-probabilities and gradients. As a result, the tanh nonlinearity does not affect the score-function gradients or the conditioning of $H$, and therefore does not directly impact the advantage transformation.
>
> That said, we agree that the advantage transformation itself can be sensitive to conditioning. This is precisely why Tikhonov damping and gradient norm clipping are important in practice; our ablations show that clipping is essential for stable training. We will clarify both the role of action squashing and these stabilization mechanisms in the revision.
>
> ---
>
> We appreciate your positive feedback on clarity and empirical results.

---

> > ### Author Rebuttal · Reviewer_x9nY · 2026-04-01
> >
> > Thank the author for the detailed reply. I have adjusted my score accordingly.

---

### Official Review · Reviewer_PQcr · 2026-03-13

**Soundness:** 2
**Presentation:** 3
**Significance:** 2
**Originality:** 3
**Overall Recommendation:** 5
**Confidence:** 4

**Summary:**

This paper develops an approximation algorithm for computing natural policy gradient (NPG). Since NPG is the solution to a weighted least-square problem, this work applies the randomized block Kaczmarz method to compute it, and the resultant formula can be viewed as using a modified advantage (13). The proposed algorithm, Randomized Advantage Transformation (RAT), is evaluated on standard RL environments in OpenAI Gym. The results show that RAT can outperform three baselines that also compute NPG.

**Compliance With Llm Reviewing Policy:**

Affirmed.

**Final Justification:**

The rebuttal have addressed most of my concerns so I increased my score.

**Key Questions For Authors:**

1. Can you provide more explanation on how (11) is derived? See the weaknesses above for more details.

2. What’s the time complexity compared to other baselines, in theory and in the experiments?

**Limitations:**

Yes

**Strengths And Weaknesses:**

*Soundness:*

- Strength: The experiments on both discrete and continuous control problems show that RAT is effective.
- Weaknesses
    - The algorithm derivation needs further discussion, especially for Eq.(11). It was motivated to solve (5) but (5) is not a least-square problem as it is regularized. (11) is not a traditional proximal problem either. Moreover, the $\Sigma$ is removed due to sampling according to the paper. However, is there a justification for it? It would be helpful to see a derivation for this removal. It is unclear how the weight for the loss can be translated to the weight of sampling rows. In summary, it is unclear what the original objective is for the update (11). This problem is critical as it also dictates the correctness of the theoretical analysis in Sec.4.3.
    - The algorithm in Sec.4.3 is used to iteratively refine the gradient and the goal is to get the NPG for the current policy parameters. Then it should be used only when it is close to convergence. This means Line 10 of Algo.1 should be outside of the inner loop over batches, but now it is inside so the applied gradient may not be accurate. The behaviour of the update is then unknown.

*Presentation:*

- Strength: Overall, the paper is well-written and easy to follow
- Weakness: Some concepts were introduced for no obvious reason. For example, what’s the purpose of Sec.4.2 when Sec.4.3 has a completely different advantage transformation?

*Significance:*

- Weakness: Computing NPG accurately is an important research topic and the new algorithm is effective. However, there is no running time comparison in both theory and experiments. This is insufficient since it will be critical to know if the performance is gained with a significant computation cost.

*Originality:*

- Strength: The paper applies the randomized block Kaczmarz method to compute NPG, which is novel and can provide more insight into the field.

Minor comment: (L149 left) $d$ should be $n$

---

> ### Author Rebuttal · Authors · 2026-03-28
>
> We thank the reviewer for the detailed and constructive feedback. We address each concern below and will incorporate the suggested clarifications in the revision.
>
> ---
>
> ### 1. Derivation of Eq. (11) and removal of $\Sigma$
> We agree this step was not sufficiently explained.
>
> Starting from the regularized least-squares formulation:
> $$g^* = \arg\min_g ||y - Hg||^2_{\Sigma} + \lambda ||g||^2_2$$
> Sampling $(s, a)\sim d_\pi(s)\pi(a|s)$ implicitly incorporates $\Sigma$ into the data distribution, yielding:
> $$E_\tau [ ||y_\tau - H_\tau g||^2_2 ]\propto ||y - Hg||^2_{\Sigma}$$
>
> Thus, $\Sigma$ does not appear explicitly in the sampled objective because it is absorbed through the sampling distribution.
>
> Eq. (11) corresponds to a **regularized block update** in the randomized block Kaczmarz framework. In the classical (unregularized) formulation, each step projects the current iterate onto the solution space of the sampled linear system $H_{\tau_j}g = y_{\tau_j}$. However, enforcing this constraint directly can be unstable in our setting due to noise and potential rank deficiency of minibatches.
>
> Following the regularized variant (e.g., Eq. (29) in Goldshlager et al., 2024), we instead use a **Tikhonov-regularized update**:
> $$g_j = \arg\min_g ||y_{\tau_j} - H_{\tau_j}g||^2_2+ \lambda ||g - g_{j-1}||^2_2$$
> which can be interpreted as a **soft projection** onto the sampled system. The regularization ensures well-posedness even when  $H_{\tau_j}$ is rank-deficient. The closed-form solution is:
> $$g_j = g_{j-1}  + (H_{\tau_j}^\top H_{\tau_j} + \lambda I)^{-1}H_{\tau_j}^\top (y_{\tau_j} – H_{\tau_j}g_{j-1})$$
> which can be rewritten using the identity
> $$(H^\top H + \lambda I)^{-1} H^\top = H^\top (\lambda I + H H^\top)^{-1}$$
> yielding Eq. (12).
>
>
> We will add this derivation in the revision and clarify that Sec. 4.2 defines the **full-batch target**, while Sec. 4.3 introduces the **randomized estimator** used in RAT.
>
> ---
>
> ### 2. Theory vs. algorithm
> We agree this distinction should be clarified. The theorems analyze RAT as a **fixed-policy linear system solver** (i.e., fixed $H, y$), while Algorithm 1 interleaves these updates with policy optimization, resulting in a **time-varying sequence of least-squares problems**.
>
> We provide a formal argument (see response to Reviewer GhTJ) showing that RAT can be interpreted as a **contractive solver tracking a slowly varying sequence of LS solutions**. The tracking error is controlled by the contraction rate from the fixed-system analysis and the magnitude of policy updates. Under our algorithm setting (small learning rates and gradient clipping, which we already employ), this yields a bounded steady-state error of order
> $$O(\max_t ||\theta_{t+1} - \theta_t||)$$
>
> We will revise the paper to explicitly state this interpretation.
>
> ---
>
> ### 3. Time complexity and PPO baselines
> The paper already reports per-update wall-clock time for RAT, FVP+CG, KFAC, and Sophia. RAT is significantly faster than CG-based methods and avoids architecture-specific approximations.
> We have additionally measured PPO runtime. As expected, PPO is faster per update (≈3–5 ms vs. ≈10–20 ms for RAT), since PPO is a purely first-order method, while RAT requires per-sample gradients and a small linear solve.
>
> Despite the higher per-update cost, RAT achieves consistently better or comparable performance, particularly on more challenging settings:
>
> + Separate AC (Ant, Humanoid): RAT significantly outperforms PPO (e.g., +14% on Ant, +48% on Humanoid).
> + Shared AC (Ant): RAT substantially outperforms PPO (≈2× higher returns).
> + Shared AC (Humanoid): RAT achieves performance comparable to PPO.
>
> These results indicate that RAT provides a favorable computation–performance trade-off, especially in settings where curvature information is beneficial.
>
> We will include PPO runtime and performance comparisons in the revision to better contextualize RAT relative to strong first-order baselines.

---

> > ### Author Rebuttal · Reviewer_PQcr · 2026-04-03
> >
> > I thank the authors for the reply and discussion. Most of my concerns have been addressed, so I increased my score. I encourage the authors to add the derivation, analysis and more complete empirical results (both plots and tables) to the revised paper.

---

### Official Review · Reviewer_GhTJ · 2026-03-15

**Soundness:** 2
**Presentation:** 3
**Significance:** 3
**Originality:** 2
**Overall Recommendation:** 4
**Confidence:** 3

**Summary:**

This paper addresses the computational cost of natural policy gradients (NPG)
in deep reinforcement learning. The standard NPG requires inverting or solving
linear systems involving the Fisher information matrix, which is expensive for
large policy networks.
Building on Woodbury-based reformulations of Tikhonov-regularized natural
gradients explored in recent concurrent work (Wu et al., 2024; Guzm´an-Cordero
et al., 2025), the authors recast the Fisher inverse as a transformation of the
advantage function, so that the natural policy gradient takes the form of a
vanilla policy gradient with a modified advantage. This reduces the problem
from inverting a parameter-dimensional matrix to solving a minibatch-sized
linear system.
Building on this reformulation, the paper proposes Randomized Advantage
Transformation (RAT), which approximates the transformed advantage iter-
atively using randomized block Kaczmarz updates over minibatches within a
single on-policy rollout. The method replaces the conjugate gradient inner
loop used in prior work (e.g., TRPO) with Kaczmarz iterations, where each
step requires only a B×B matrix solve and a standard backpropagation pass
through a PPO-like surrogate loss, rather than a full-batch Fisher-vector prod-
uct. The method requires no explicit Fisher construction and no architecture-
specific factorizations like KFAC. The paper also describes how to handle shared
actor-critic networks via a pseudo-advantage for the critic.
Two convergence theorems are provided for the Kaczmarz iteration under ide-
alized (exact advantage) and noisy settings.
Experiments span MuJoCo continuous control (with both separate and shared
actor-critic networks) and Procgen visual control with ResNet policies, compar-
ing against FVP+CG, KFAC, ACKTR, PPO, and a diagonal Fisher baseline
(Sophia). RAT matches or exceeds baselines across most settings.

**Compliance With Llm Reviewing Policy:**

Affirmed.

**Final Justification:**

I thank the authors for the rebuttal. The tracking-error interpretation is a reasonable heuristic justification, though informal: the bound depends on an unknown Lipschitz constant controlled only indirectly via clipping and small learning rates. I accept this as honest motivation rather than a formal guarantee. The PPO comparisons show RAT is ~3× slower per update with environment-dependent gains. The shared AC architecture-agnosticism argument is convincing. I maintain 4 (weak accept): a practical recipe that is somewhat incremental but genuinely useful, especially for shared actor-critic settings.

**Key Questions For Authors:**

1. Theory-practice gap in Algorithm 1. Theorems 1 and 2 analyze
convergence of the Kaczmarz iteration on a fixed linear system, but Algo-
rithm 1 updates θafter every minibatch (line 10), making the system non-
stationary across inner iterations. Do the authors view the convergence
analysis as directly characterizing Algorithm 1, or as motivation for the
algorithmic design? If the former, could the authors explain why updating
θ at each inner step (which changes the importance ratio and therefore
the surrogate gradient) does not invalidate the fixed-system assumptions?
If the latter, could the authors discuss this gap explicitly? Additionally,
Lemma 2 and Theorem 3 assume parameters in [0,1]—could the authors
justify this assumption in the RL context? A satisfactory response—either
a formal argument that the gap is controlled, or an honest reframing of
the theorems as motivational—would significantly improve my assessment
of soundness.
2. Distinction from Guzm´an-Cordero et al. (2025). The paper states
that RAT differs from prior Woodbury-based approaches by transforming
the advantage rather than directly approximating the inverse Fisher. At
the algebraic level, these appear equivalent: both require solving a system
involving HτH⊤
τ , and the difference seems to lie in whether one multiplies
by H⊤ explicitly or lets backpropagation handle it. Could the authors
clarify what RAT enables that the approach of Guzm´an-Cordero et al.
(2025) cannot? A concrete answer—e.g., a setting where the surrogate-
loss formulation is necessary, or a quantitative comparison—would help
me better assess originality.
3. Wall-time comparison against PPO. RAT requires per-sample gra-
dients and a B×B matrix solve per minibatch, which adds cost relative
to PPO. Could the authors provide wall-time comparisons against PPO,
ideally on the MuJoCo tasks where RAT shows the largest return im-
provements (Ant, Humanoid)? This would clarify whether RAT’s sample
efficiency gains translate to practical speedups or whether the per-step
overhead negates them. This comparison would meaningfully affect my
assessment of significance.
4. PPO baseline in Figure 3. PPO is included in the shared-network
experiments (Table 1) but omitted from the separate actor-critic experi-
ments (Figure 3). Since PPO is the most widely used practical baseline
in this setting, could the authors include it in Figure 3? Understanding
how RAT compares to PPO with separate networks would help me assess
whether the gains over FVP+CG and KFAC reflect a genuine advantage
or whether PPO already achieves comparable returns more cheaply.
5. Wall-clock times for shared actor-critic networks. Given that
shared-network support is presented as a key contribution, could the au-
thors provide wall-clock time per update for the shared-network experi-
ments (Table 1)?

**Limitations:**

yes

**Strengths And Weaknesses:**

1.3 Strengths
1. Clean practical recipe with broad applicability. The core method
is simple and appealing: transform the advantage via a B×B solve, then
backpropagate through a PPO-like surrogate loss. This avoids the con-
jugate gradient inner loop of TRPO-style methods and the architecture-
specific Kronecker factorizations of KFAC. The method is genuinely architecture-
agnostic—it works with MLPs, ResNets, and shared actor-critic networks
out of the box, which is a meaningful practical advantage as RL moves
toward more diverse architectures.
2. Solid empirical coverage. The experiments span three distinct settings:
MuJoCo with separate actor-critic networks, MuJoCo with shared net-
works, and Procgen with ResNet policies. RAT matches or outperforms
all baselines in most settings, with particularly clear gains on challeng-
ing tasks (Ant, Humanoid). The ablation study (Figure 5) is thorough
and shows robustness to batch size, number of Kaczmarz iterations, and
damping coefficient, suggesting the method does not require fine-grained
tuning.
3. Useful conceptual reframing. The observation that natural policy
gradients can be expressed as vanilla policy gradients with a curvature-
corrected advantage is pedagogically valuable. Even if the underlying
2
Woodbury algebra is known, packaging it as “modify the advantage, then
do standard policy gradient” provides a clean way to think about what
natural gradients accomplish and makes the method easy to implement in
existing codebases.
4. Support for shared actor-critic networks. Shared-network settings
are practically important but poorly served by existing NPG methods
(FVP+CG is not directly applicable; KFAC requires nontrivial exten-
sions). RAT handles this via a pseudo-advantage for the critic (Ap-
pendix C.1), and the shared-network results (Table 1) show clear improve-
ments over ACKTR and PPO on several tasks.
1.4 Weaknesses
1. The convergence theory does not describe the implemented algo-
rithm. Theorems 1 and 2 analyze convergence of the Kaczmarz iteration
on a fixed linear system: the data matrix H, advantage vector y, and
regularization parameter λ are all held constant across inner iterations,
and the iterate gj converges to the solution g∗of the unperturbed system.
However, Algorithm 1 updates θ after every minibatch (line 10), which
changes the policy and therefore the surrogate loss gradient at line 8
(via the importance ratio πθ/πold). This makes the linear system non-
stationary across inner iterations. The resulting algorithm is closer to
“multiple PPO-like updates per rollout with curvature-corrected advan-
tages” than to an iterative linear solver converging to a fixed target. This
gap is not acknowledged anywhere in the paper. Additionally, Lemma 2
and Theorem 3 (linear convergence of RAT, Appendix lines 751 and 784)
assume parameters lie in [0,1], and it is not clear why this assumption is
justified in the RL setting. I would welcome clarification from the authors
on whether the convergence analysis is intended as a direct characteriza-
tion of Algorithm 1 or as motivation for the design.
2. Missing wall-time comparison against PPO. Table 2 reports wall-
clock time per update for RAT, FVP+CG, KFAC, and Sophia, but omits
PPO—the most widely used baseline and the most important practical
competitor. RAT requires per-sample gradients and a B×B matrix solve
per minibatch, which is certainly more expensive per step than PPO’s
clipped surrogate. Without a wall-time comparison, the most practically
relevant question—whether RAT’s improved sample efficiency compen-
sates for its higher per-step cost—cannot be answered. PPO is also miss-
ing from the separate actor-critic MuJoCo experiments (Figure 3), making
it harder to assess RAT’s value in that setting. Furthermore, wall-clock
times for the shared actor-critic setting are absent entirely, despite shared-
network support being presented as a key contribution.
3. Relationship to concurrent work could be clarified. The Woodbury
reformulation of Tikhonov-regularized natural gradients appears in Wu et
3
al. (2024) and Guzm´an-Cordero et al. (2025). The paper states that it dif-
fers by transforming the advantage rather than directly approximating the
inverse Fisher, but it is not entirely clear to me how deep this distinction
runs—at the algebraic level, these appear to be equivalent operations, with
the difference lying primarily in implementation (backpropagation through
a surrogate loss vs. explicit gradient computation). I would welcome clar-
ification from the authors on what specifically cannot be achieved by the
approach of Guzm´an-Cordero et al. (2025) that RAT enables. I do see
value in the practical recipe the paper offers—architecture agnosticism,
shared actor-critic support, and simplicity of implementation are genuine
merits. But the presentation may somewhat overstate the conceptual nov-
elty relative to what is, in my reading, a careful and effective combination
of known components. Similarly, the distinction from SPRING (Gold-
shlager et al., 2024) rests on RAT performing inner iterations within a
rollout rather than one update per batch, but given the theory-practice
gap noted in W1, this distinction is less principled than presented: the
inner iterations update θat each step rather than refining a fixed estimate.

---

> ### Author Rebuttal · Authors · 2026-03-28
>
> We thank the reviewer for the detailed and constructive feedback. We address each concern below and will incorporate the suggested clarifications in the revision.
>
> ---
>
> ### 1. Theory vs. algorithm
>
> We do not claim the theorems directly characterize Algorithm 1, but rather provide intuition for its behavior. The theorems analyze RAT as a fixed-policy linear system solver, while Algorithm 1 interleaves these updates with policy optimization, resulting in a time-varying sequence of systems.
>
> To clarify the gap formally, let $g_t^ * $ denote the solution of the regularized least-squares problem defined by the current policy $\theta_t$, and define the tracking error $e_t:=||g_t - g_t^ * ||$. At iteration t, RAT performs an update yielding $g_{t+1}$. The fixed-system analysis (Theorem 1) implies a contraction:
>
> $||g_{t+1} - g_t^ * || \leq \rho ||g_t - g_t^ * || = \rho e_t$, $\rho := 1-\mu <1$.
>
> After the policy update $\theta_t\mapsto\theta_{t+1}$, the target solution shifts. Under standard smoothness assumptions on $H(\theta)$ and $y(\theta)$, the solution map $\theta\mapsto g^*(\theta)$ is Lipschitz:
>
> $||g^ * (\theta_{t+1}) - g^*(\theta_t)|| \leq L||\theta_{t+1} - \theta_{t}||$
>
> Combining these yields:
>
> $e_{t+1}\leq \rho e_t + L||\theta_{t+1} - \theta_t||$
>
> Unrolling:
>
> $e_t \leq \rho^t e_0 + L\sum_{s=0}^{t-1}\rho^{t-1-s}||\theta_{s+1} - \theta_s||$
>
> This shows that RAT can be interpreted as a **contractive solver tracking a slowly varying sequence of systems**. Under our settings (small learning rates and gradient clipping), the drift term remains small, yielding a bounded steady-state error of order
>
> $O(\max_t ||\theta_{t+1} - \theta_t||)$
>
> This aligns with standard analyses of stochastic approximation in RL, where updates track a moving target induced by policy changes, and provides a heuristic justification for the algorithm.
> We will revise the paper to clarify this tracking-error interpretation.
>
> ---
>
> **Regarding the appendix assumption.**
>
> We agree this point was unclear. The boundedness assumption appears only in an auxiliary appendix result and is not intended as a claim that RL policy parameters lie in $[0, 1]$. The confusion likely arises from notation misuse, where $\theta$ was overloaded in that argument. We will revise the appendix to correct the notation and clarify the role of this assumption.
>
> ---
>
> ### 2. Relation to Woodbury
>
> We agree that the Woodbury reformulation itself is related to recent concurrent work. Our claim is that RAT is the first method to express natural policy gradients purely as a standard loss, enabling computation via backprop without explicit Fisher.
>
> A setting where this distinction is particularly important is the **shared actor-critic architecture**. While Woodbury-based approaches (e.g., Guzmán-Cordero et al., 2025) can in principle be applied in this setting, they typically require maintaining separate curvature-adjusted gradients for actor and critic and carefully merging them during parameter updates. This merging is inherently **architecture-dependent**, as it requires explicit knowledge of which parameters are shared and how gradients from different heads should be combined.
>
> In contrast, RAT introduces a **pseudo-advantage formulation** that unifies actor & critic objectives into a single surrogate loss, whose gradient is computed via standard backprop. As a result, curvature-adjusted updates are handled implicitly by autograd, without requiring manual gradient partitioning or architecture-specific merging logic. This allows RAT to remain **architecture-agnostic in practice**, even in shared-network settings.
>
> ---
>
> ### 3. Runtime, PPO comparison, and timing
> We thank the reviewer for raising this important point. We measured PPO runtime and, as expected, PPO is faster per update (≈3–5 ms vs. ≈10–20 ms for RAT), since PPO is a purely first-order method, while RAT requires per-sample gradients and a small linear solve.
>
> We report the timing and performance comparison below:
>
> **Separate AC**
> * Ant:
> * * PPO: 4707.8±117.8(returns); 3.18±1.36ms
> * * RAT: **5376.7±98.3(returns)**; 10.04±1.35ms
> * Humanoid:
> * * PPO: 4553.4±849.3(returns); 3.22±1.40ms
> * * RAT: **6761.8±79.3(returns)**; 18.17±3.04ms
>
> **Shared AC**
> * Ant:
> * * PPO: 1373.9±26.0(returns); 4.71±1.46ms
> * * RAT: **2926.6±353.1(returns)**; 11.66±1.55ms
> * Humanoid:
> * * PPO: 5357.9±150.9(returns); 4.72±1.49ms
> * * RAT: **5382.7±117.3(returns)**; 19.85±3.11ms
>
> Despite higher per-update cost, RAT is most beneficial in regimes where curvature matters (e.g., high-dimensional settings), where PPO often plateaus or requires careful tuning. RAT is not designed to match PPO’s per-step efficiency, but to provide a simple, architecture-agnostic, and principled approximation to natural policy gradients. Compared to existing natural-gradient methods, it offers a stronger performance–compute trade-off while avoiding architecture-specific approximations and complex inner solvers.
>
> We will include these comparisons in the revision.

---

> > ### Author Rebuttal · Reviewer_GhTJ · 2026-04-07
> >
> > Acknowledgement: (b) Partially resolved - I have follow-up questions for the authors.
> >
> > Reasons:
> >
> > I thank the authors for their rebuttal. The authors have addressed the majority of my concerns.
> > The PPO wall-time and return comparisons resolve my earlier question: the ~3-5x per-update overhead is now transparent and readers can judge the tradeoff for themselves.
> > My score remains the same.
> >
> > One follow-up:
> >
> > 1. **Theory-practice gap (W1).** The tracking-error reframing, RAT as a contractive solver on a slowly varying system, is more honest than the original presentation and I find it reasonable. That said, the bound's practical content depends on the drift term $L\|\theta_{t+1} - \theta_t\|$ remaining small. Could the authors report this quantity empirically (e.g., on Ant or Humanoid across training) to verify that the tracking regime actually holds? Without this, the argument is plausible but unverified.

---

> > > ### Author Response · Authors · 2026-04-07
> > >
> > > We thank the reviewer for this suggestion. In the tracking bound, the relevant drift contribution is $L|\theta_{t+1} - \theta_t|$. While the Lipschitz constant $L$ is not directly measurable, the step size $|\theta_{t+1} - \theta_t|$ is explicitly controlled in our implementation. In particular, we use $l_2$ gradient norm clipping at $0.5$ together with a learning rate of $0.05$, so each parameter update is bounded by at most $0.5 \times 0.05 = 2.5 \times 10^{-2}$ in norm. This keeps the drift term small throughout training up to the unknown constant $L$.
> > >
> > > More broadly, this is precisely why gradient clipping and small learning rates are important in our implementation: they ensure that the target system evolves slowly enough for the tracking interpretation to be meaningful. We will clarify this point in the revision.

---

### Decision · Program_Chairs · 2026-04-30

**Decision:**

Accept (regular)

**Comment:**

The reviewers find this to be a technically solid and useful paper on natural policy gradients. The main strengths are the clean and practical formulation of RAT, its architecture-agnostic implementation via standard backpropagation, strong empirical results across MuJoCo and Procgen, and support for shared actor-critic architectures. Multiple reviewers also valued the theoretical analysis and the paper’s clarity, and the rebuttal was effective in addressing most concerns. In particular, the added explanation of the derivation, the clarification of the relationship to prior Woodbury-based methods, and the new PPO runtime/performance comparisons substantially strengthened the submission.

The main remaining concerns are about the gap between the fixed-system convergence theory and the implemented algorithm, as well as the degree of conceptual novelty relative to concurrent work. While the authors’ tracking-error interpretation and clarifications are reasonable, this aspect remains more heuristic than fully verified, and should be stated carefully in the final version. The revision should also incorporate the promised derivational details, runtime comparisons, clearer discussion of the fixed training budget, and a more precise positioning against related methods. Overall, the reviewer consensus is positive, and I lean toward acceptance.